# Mid infrared gas spectroscopy using efficient fiber laser driven photonic chip-based supercontinuum

Davide Grassani [1], Eirini Tagkoudi [1], Hairun Guo [2,3], Clemens Herkommer[2,4], Fan Yang [5], Tobias J. Kippenberg[2] & Camille-Sophie Brès[1]

Directly accessing the middle infrared, the molecular functional group spectral region, via supercontinuum generation processes based on turn-key fiber lasers offers the undeniable advantage of simplicity and robustness. Recently, the assessment of the coherence of the mid-IR dispersive wave in silicon nitride ($Si_3N_4$) waveguides, pumped at telecom wavelength, established an important first step towards mid-IR frequency comb generation based on such compact systems. Yet, the spectral reach and efficiency still fall short for practical implementation. Here, we experimentally demonstrate that large cross-section $Si_3N_4$ waveguides pumped with 2 μm fs-fiber laser can reach the important spectroscopic spectral region in the 3–4 μm range, with up to 35% power conversion and milliwatt-level output powers. As a proof of principle, we use this source for detection of $C_2H_2$ by absorption spectroscopy. Such result makes these sources suitable candidate for compact, chip-integrated spectroscopic and sensing applications.

[1] Ecole Polytechnique Fédérale de Lausanne, Photonic Systems Laboratory (PHOSL), STI-IEL, Station 11, Lausanne CH-1015, Switzerland. [2] Ecole Polytechnique Fédérale de Lausanne, Laboratory of Photonics and Quantum Measurements (LPQM), SB-IPHYS, Station 3, Lausanne CH-1015, Switzerland. [3] Key Laboratory of Specialty Fiber Optics and Optical Access Networks, Shanghai University, Shanghai 200343, China. [4] Physics Department, Technical University of Munich, Garching 85748, Germany. [5] Ecole Polytechnique Fédérale de Lausanne, Group for Fibre Optics (GFO), STI-IEL, Station 11, Lausanne CH-1015, Switzerland. Correspondence and requests for materials should be addressed to D.G. (email: davide.grassani@epfl.ch) or to C.-S.B. (email: camille.bres@epfl.ch)

The middle infrared (mid-IR) spectral region (2–10 μm) has a high technological importance for spectroscopy and sensing with applications in health and environmental monitoring[1,2]. Particularly, the 3–5 μm region, the functional group region, is interesting as it hosts the first mid-IR atmospheric window, and contains the molecular fingerprint of many hydrocarbons, nitrogen dioxide, greenhouse gases, and specimens detectable in breath analysis[3]. Besides direct mid-IR generation with quantum cascade lasers[4], interband cascade lasers[5], and $Fe^{2+}$ doped crystals[6], wavelength conversion in nonlinear materials is a promising solution to reach the mid-IR spectral range. Very efficient conversion towards mid-IR has been demonstrated through frequency-divide-by-two optical parametric oscillators[7,8] and difference frequency generation (DFG) between two telecom band signals[9–13].

Broadband mid-IR generation can also be achieved by supercontinuum generation (SCG) in soft glass optical fibers[14–23] or photonic integrated waveguides[24–31]. Compared to other wavelength conversion schemes, SCG offers some advantages[32]: it is in fact a compact single-pass geometry that does not require any additional seed laser or temporal synchronization, and it can provide a broader and tunable mid-IR emission. When CMOS compatible materials are employed, SCG platforms benefit from lithographic precision and high yield, and usually have low power consumption. Recent works have shown that such approach can be applied as well to dual-comb spectroscopy[32].

Often, to extend the reach into the mid-IR, the nonlinear waveguides are pumped with an optical parametric oscillator placed beyond the 2-micron wavelength range, but there is interest in driving such platforms with femtosecond mode-locked fiber lasers, which are reliable, easy to use, and compact frequency comb sources.

However, up to now, very few demonstrations succeeded in generating supercontinua following this paradigm. High energy (2–3 nJ) femtosecond pulses in the 2-micron region can lead to more than 30% of conversion efficiency (CE) beyond 2200 nm in an $InF_3$ fiber[22], and spectral broadening extending beyond 4.5 μm in Telluride photonic crystal fibers[23]. Despite the large CE and spectral extent, the coherence of such mid-IR emission has not been experimentally investigated. Conversely, low pulse energy (18 pJ) at 2000 nm can lead to coherent mid-IR SCG in chalcogenides nanospikes[16,17] or nanotapers[19], but early damage power threshold and high mid-IR absorption from the cladding avoid power scaling up to the milliwatt level. In terms of chip integrated waveguides, mid-IR extended SCG has been experimentally demonstrated in AlN waveguides pumped with 0.8 nJ from a telecom band femtosecond mode-locked fiber laser[24]. The power generated inside the waveguide in the 3000–4000 nm range is about 0.3 mW, corresponding to a CE close to 0.5%. Overall, none of the above-mentioned approaches has been employed for mid-IR spectroscopy demonstrations.

Also, in the last years there has been a trend in miniaturizing and simplifying DFG schemes by utilizing single pump configuration with chip-scale nonlinear platforms. Emission up to 5.5 μm was obtained by DFG between the pump and the long-wavelength dispersive wave (DW) in AlN waveguides[24], while intrapulse DFG, generated from an Er:doped fiber mode-locked laser, enables tunable mid-IR radiation in the 4–5 μm region using PPLN waveguides[33]. However, efficiencies are limited to below 0.5%.

Recently, we showed that direct generation of mid-IR light, from an erbium-doped fiber laser at 1.56 μm, is possible through DW generated in $Si_3N_4$, and asserted the phase coherence and frequency comb nature[34]. This platform has the potential to merge all the desired features addressed separately in the above-mentioned devices: a power scalable, fiber laser pumped coherent mid-IR generation in a low loss chip-scale waveguide with lithographical control of its dispersion.

However, the outstanding problem in this first demonstration was the extremely low CE and insufficient power beyond 3 μm for future molecular fingerprinting. Indeed, reaching efficient DW generation beyond 3 μm is still difficult in CMOS platforms directly pumped by fiber lasers. The larger is the spectral coverage, the lower is the power transferred in the targeted region. Moreover, the SCG process can convert a non-negligible portion of the pump energy over unwanted spectral bands, such as in the visible, further decreasing the CE in the region of interest[35].

In this work, we overcome these hurdles and demonstrate a turn-key, high-efficient, and compact mid-IR source based on DW generation with power levels sufficient for spectroscopy application. We leverage both recent advances in fiber laser technologies, which allow shifting their emission wavelength in the short wave infrared (SWIR), until the limit of the silica absorption edge (around 2.1 μm), and large cross-section waveguide designs[36], enabling considerable freedom in dispersion engineering and low mid-IR propagation losses. In this way we efficiently convert, to the 3–4 μm wavelength region, a commercial SWIR femtosecond fiber laser by setting the pump wavelength as to favor dispersion for targeted long wavelength operation. Record CE, defined as on-chip generated mid-IR DW power over on-chip coupled pump laser power, as high as 35% is reached at 3.05 μm, and close to 20% at 3.950 μm, corresponding to more than one milliwatt average power at the chip output. A systematic experimental and numerical study on the generation of mid-IR DW provides new information on the efficiency and dynamics of the DW generation process as a function of the pump power. Finally, as a proof-of-principle, we successfully exploit this on-chip mid-IR source for $C_2H_2$ detection through gas absorption spectroscopy.

## Results

**Design of mid-IR source through dispersive wave optimization.** It is well known that propagation of sufficiently powerful femtosecond laser pulses in the anomalous group velocity dispersion (GVD) region of a nonlinear waveguide can induce high order soliton dynamics[35], leading to an initial spectral broadening caused by self-phase modulation (SPM) and subsequent temporal compression which are proportional to the soliton number[37]. At the compression point, when the spectrum is extensively broadened, the soliton can be perturbed by high order dispersion (HOD) or nonlinear terms resulting in soliton fission, and can transfer energy to linear DW spectrally shifted from the pump[35]. The DW generation process can occur at the frequency where the phase constant of the soliton pulse equals the one of the linear wave[38], and is thus given by the phase matching condition:

$$\beta(\omega) - \beta(\omega_s) - v_g^{-1}(\omega - \omega_s) = \frac{\gamma P}{2} \qquad (1)$$

where $\beta$ is the mode propagation vector, $\omega_s$ is the soliton central frequency, $v_g$ is the soliton group velocity, $P$ is the pulse peak power, and $\gamma$ is the waveguide's effective nonlinearity. The nonlinear phase shift on the right-hand side of Eq. (1) is small and is usually neglected. The left-hand side of Eq. (1) is called integrated dispersion $\beta_{int}$, and can be rewritten as a Taylor expansion leading to

$$\beta_{int} = \sum_{k \geq 2} \frac{(\omega - \omega_s)^k \partial^k \beta}{k!} \bigg|_{\omega = \omega_s} \approx 0 \qquad (2)$$

The GVD determines the location of the phase matching points and it has been shown that the generation of a phase matched DW corresponds to the occurrence of a zero dispersion

wavelength (ZDW)[39,40]. Also, HOD terms ($k > 2$) affect the amount of power transfer to the DW[38,40,41]. Qualitatively, even-order dispersion terms lead to two DWs with symmetric intensity and frequency detuning[42] with respect to the pump, while positive or negative odd-order terms break this symmetry favoring blue or red-shifted DW, respectively[40]. The material dispersion of $Si_3N_4$ can easily lead to a first ZDW point in the near-IR for the fundamental waveguide mode, responsible for DW generation at visible wavelengths. The large anomalous material GVD in the mid-IR has to be compensated by waveguide dispersion in order to reach a second ZDW at longer wavelengths. However, efficiently converting the light to the mid-IR DW means that conversion towards the unwanted visible DW has to be limited, and mid-IR mode confinement has to be improved to avoid cladding absorption. These limitations can be mitigated, as seen in Fig. 1, by combining large cross-section waveguides, which satisfy both the necessary confinement and dispersion

engineering, with accurate positioning of the pump wavelength, to meet the requirements on HOD terms. The waveguides we use in this study have a height that can reach 2.2 μm, with width in the 1 μm range, while the central wavelength of the pump laser can be tuned between 2070 and 2090 nm.

In Fig. 1a, we show the computed GVD of standard (with a height around 870 nm) and large cross-section $Si_3N_4$ waveguides. The second ZDW of the latter waveguides is further red-shifted compared to standard ones, enabling the generation of DW deeper in the mid-IR for the same pump wavelength. At the same time, the large cross-section waveguide can significantly reduce mid-IR absorption in the silica cladding through improved mode confinement, as seen in Fig. 1b, where we computed the absorption losses $\alpha$ by including the imaginary part of the refractive index of silica in our numerical simulations (see Methods). The amount of integrated dispersion separating the pump from the DW phase matched wavelengths, which we will refer to as dispersion barrier, clearly illustrates a symmetry breaking in DW generation. As seen in Fig. 1c, a 2090 nm pump in large cross-section waveguides generates a mid-IR DW in the same wavelength range as the one obtained by using standard waveguides with a 1560 nm pump. However, 2090 nm pumping in large waveguides leads to a much lower mid-IR dispersion barrier, and a much higher visible one, that should clearly favor mid-IR power transfer. It has to be noticed that even if the large cross-section waveguide pumped at 1560 nm in theory features a DW generated beyond 5 μm, the mid-IR dispersion barrier will also significantly increase (Fig. 1c), greatly limiting the efficiency.

**Experimental implementation and simulated dynamics**. The experimental set-up is detailed in Fig. 2a. The pump source is a commercial, turn-key soliton self-frequency shifted thulium-doped fiber mode-locked laser (NOVAE Brevity λ+), with pulse duration at full width half maximum (FWHM) of 78 fs, bandwidth of 60 nm centered at 2090 nm, repetition rate of 19 MHz and average power of about 100 mW. Before coupling to the waveguide, the polarization is managed and the power can be varied with a variable optical attenuator (VOA). Light is coupled into the fundamental transverse magnetic (TM) polarization mode of the waveguide, facilitated by inverse tapers and using two identical aspheric black diamond lenses. At the device output, the collimated light is focused by means of a parabolic mirror onto a fluoride multimode fiber (MMF) and the spectra are recorded with a Fourier Transform Optical Spectrum Analyzer (FT-OSA) spanning the 1–5 μm range (Thorlabs OSA205C). The top view of the device is acquired with a microscope objective which projects the image on a visible camera. The samples consist of 5 mm long straight $Si_3N_4$ waveguides buried in $SiO_2$. We investigated waveguides with four different nominal widths: 1000, 1050, 1100, and 1175 nm. The waveguide thickness slightly increases with larger widths, ranging from 2.09 to 2.19 μm. The total coupling losses were around 11 dB for the larger waveguides, but 1.5 and 2.5 dB higher for the 1050 and 1000 nm respectively, while propagation losses are 0.2 dB/cm. Figure 2b shows the experimentally recorded output spectrum from the 1100 nm width waveguide for a coupled average power of 13.6 mW, corresponding to about 6.8 kW peak power and coupled pulse energy of 0.75 nJ. For these values, the soliton number is around 5. A clear DW is observed at the expected phase matching position near 3.5 μm.

Indeed the obtained simulated spectrum reproduces well the experimental one after a propagation of 4.2 mm, which matches the length of the straight waveguide section, without considering the tapered regions. It also shows a weaker visible DW around 500 nm which cannot be detected by the FT-OSA, but justifies the observed green light scattered out of the chip (see Fig. 2a).

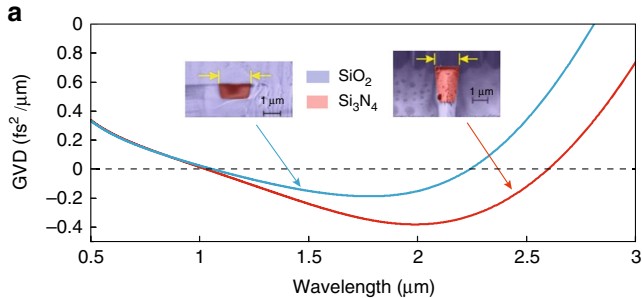

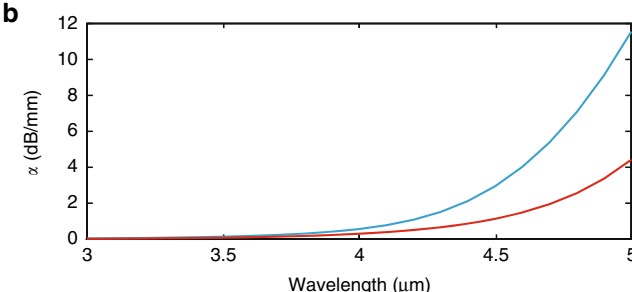

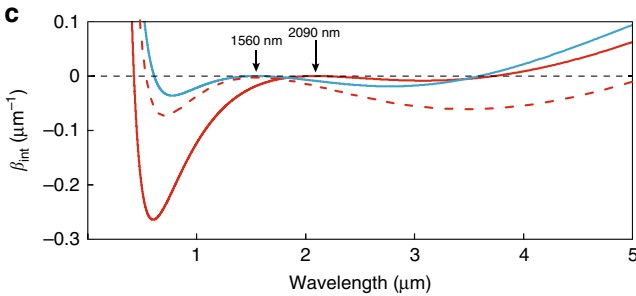

**Fig. 1** Comparison between standard and large cross-section silicon nitride ($Si_3N_4$) waveguides. **a** Group velocity dispersion (GVD) for the TE fundamental mode of a standard waveguide with cross-section 870 × 1700 nm$^2$ (blue) and the TM fundamental mode of a large cross-section waveguide with dimension 2177 × 1150 nm$^2$ (red). The insets show scanning electron microscope (SEM) images of the waveguide cross-section. **b** Attenuation coefficient $\alpha$ as a function of wavelength for (blue) standard and (red) large cross-section waveguides. **c** Solid lines: integrated dispersion as a function of the wavelength for the standard waveguide pumped at 1560 nm (blue) and the large cross-section one pumped at 2090 nm (red). The dispersive wave (DW) phase matching points lie in the same region for both configurations. Dashed red line: integrated dispersion as a function of the wavelength for the large cross-section waveguide pumped at 1560 nm. Pump positions are indicated by the arrows

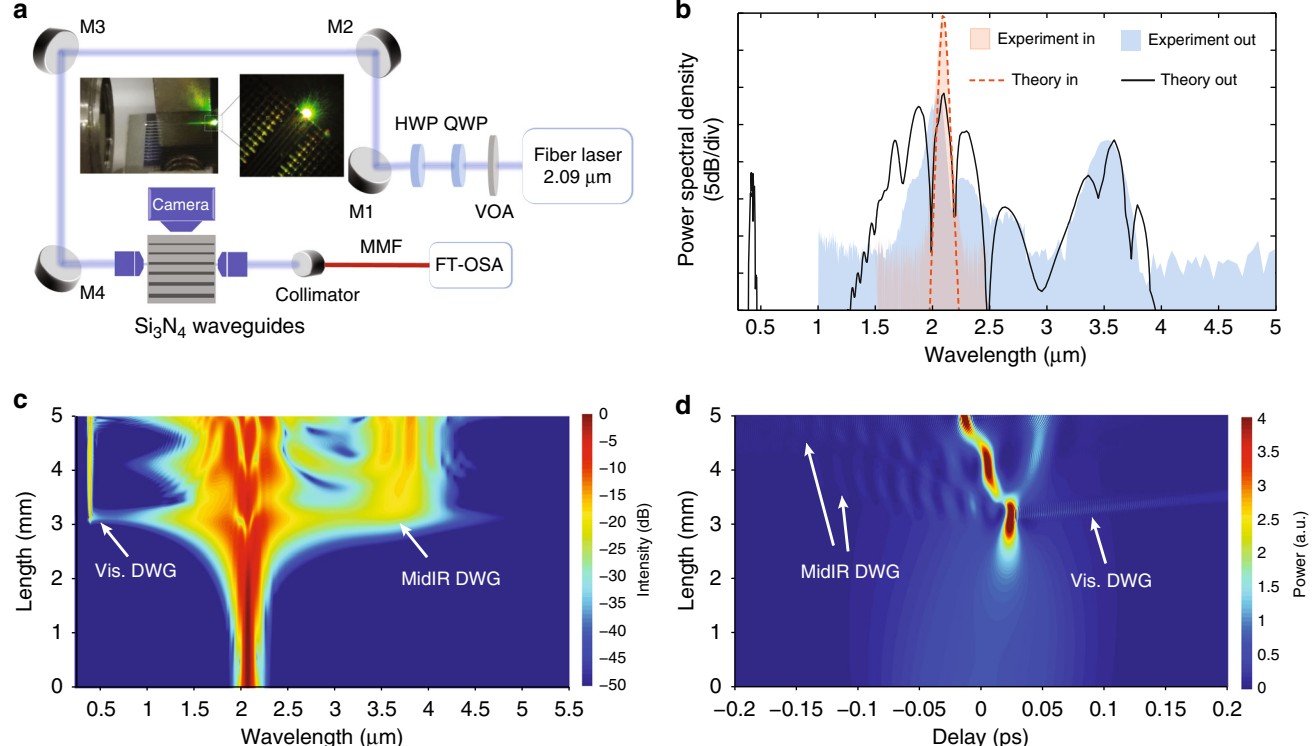

**Fig. 2** Experimental set-up and dispersive wave generation. **a** Experimental set-up. HWP half wave plate, QWP quarter wave plate, M mirror, MMF multimode fiber. The left inset is a picture of the chip and input coupling lens, while the right one is an image of the output of the waveguide taken from the top camera. Both images are taken under the same pumping condition. **b** Experimental (shaded region) and theoretical simulated (solid and dotted lines) input and output spectra for 13.6 mW average power coupled in the 1100 nm width waveguide. Spectral (**c**) and temporal (**d**) pump pulse evolution (dB scale) over the waveguide length for an input pulse of 110 fs at 2090 nm, with frequency chirp $C = -1000 \ \mathrm{fs}^2/2\pi$

Looking at the simulated pulse evolution over the waveguide length (Fig. 2c), we notice how both the visible and mid-IR DWs are generated at the soliton compression point which takes place at around $l_c = 3.5$ mm, also qualitatively confirmed on the waveguide image in Fig. 2a. Moreover, Fig. 2c shows how, after the first compression point, additional spectral broadening points separated by a much shorter distance occur. This behavior can be understood by looking at the temporal evolution in Fig. 2d. Just after the first compression point, the propagating pulse separates in two pulses in a process known as the soliton splitting effect[43]. Due to HOD and high order nonlinear effects (e.g., the self-steepening effect), one of the pulses has more energy and can undergo sufficient broadening to again overlap with the DW. The soliton self-compression process can thus repeat, leading to multiple generation of DW as long as the new pulse maintains enough energy for the necessary spectral broadening. This mechanism thus reinforces the power in the DW spectral region.

**Evolution and efficiency of DW generation**. The simulated integrated dispersions (Fig. 3a) indicate that the 2090 nm pump significantly favors mid-IR DW for the four waveguide geometries, and that the phase matching point continuously shifts towards longer wavelength with increasing width. This expected behavior is confirmed experimentally as seen in Fig. 3b. We measure a DW peak at 3050, 3220, 3530, and 3950 nm for the 1000, 1050, 1100, and 1175 nm waveguide width, respectively. There is a difference with the theoretically predicted phase matched wavelengths for the smaller waveguides, which can be primarily due to the recoil of the central soliton toward shorter wavelengths and small variation in the actual waveguide

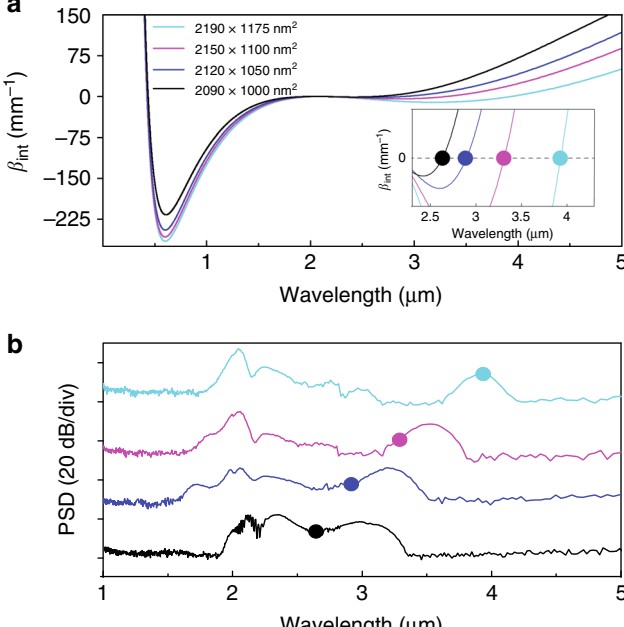

**Fig. 3** Dispersive wave generation. **a** Integrated dispersion for the four waveguides under test. The inset shows expected mid-IR phase matching points. **b** Experimental spectra recorded at an estimated average coupled power of 13.6 mW, for the four different waveguides under test (same color convention as for (**a**)). Points show the expected mid-IR phase matching points

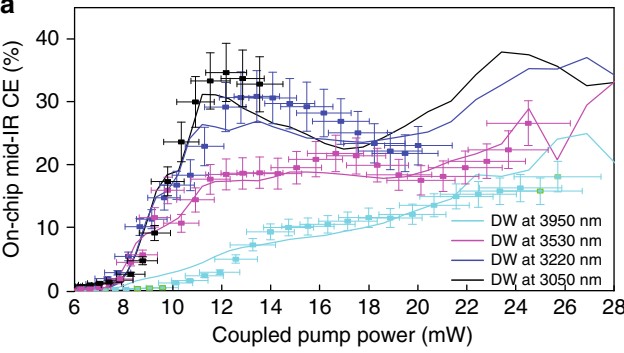

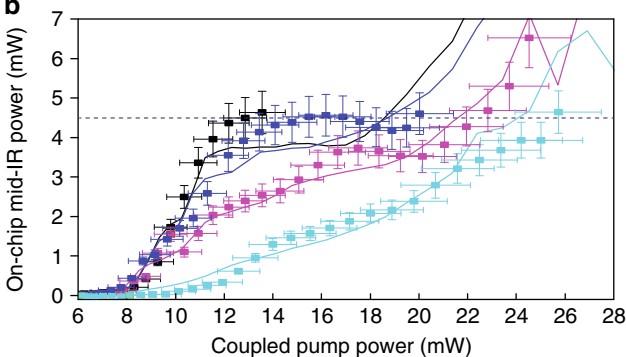

**Fig. 4** Dispersive wave efficiency. Scattered points represent experimental data, lines are simulations. **a** mid-IR DW generation on-chip conversion efficiency (CE) as a function of coupled pump power. Error bars of ±0.5 dBm and ±0.25 dBm represent the standard deviation of repeated measurements of DW and coupled pump powers, respectively. **b** On-chip mid-IR DW power estimated from measured spectra. Error bars: same as for (a). 4.5 mW of on-chip DW power is reached at 3050 nm with 13 mW pump power, at 3220 nm with 16 mW of pump power, at 3530 nm with 22 mW of pump power, and 3950 nm with 26 mW of pump power

dimensions (see Supplementary Note 1). In fact, because of the negligible Raman response[39], here we do not see the red-shift of the soliton and the consequent trapping of DW, as observed in silica fibers[44,45]. The spectra of Fig. 3b also show that, apart from some pump broadening due to SPM, most of the converted pump energy lies in the DW bandwidth. We both experimentally retrieved and numerically simulated the on-chip CE for all tested waveguides (see Methods). It should be noted that the 1000 nm wide waveguide suffered from higher input coupling loss, which limited the range of coupled pump power.

The results for the on-chip mid-IR DW CE as a function of the coupled pump power are plotted in Fig. 4a. We observe a power threshold above which the DW is generated. This threshold comes from the fact that the power must be high enough for the soliton compression point to occur before the end of the waveguide since[37] $l_c \propto (|\beta_2|P)^{-1/2}$. In addition, the broadening, which is proportional to the soliton number[37] $N \propto (P/|\beta_2|)^{-1/2}$ must be sufficient to overlap with the DW phase matched wavelength. For the two smaller waveguides, the efficiency rapidly increases with pump power, reaching record values in the 30–35%. For the larger waveguides, the efficiency does not increase as quickly, mostly limited by the larger spectral separation between pump and DW. Additionally, larger waveguides have a higher value of $|\beta_2|$ at the pump wavelength (0.32 and 0.4 fs²/μm, respectively, compared to 0.14 and 0.22 fs²/μm for the 1000 and 1050 nm wide waveguides). For a given pulse peak power and width, the soliton number is thus smaller[46], limiting both the compression factor and the number of compression

points before the end of the waveguide (see Supplementary Figure 1 for a numerical comparison between all the different waveguides). Therefore less power can be coupled to the DW. Nevertheless, CEs close to 20% are still experimentally measured. Interestingly, the CE for the two smaller waveguides starts decreasing beyond a coupled power of about 12 mW, a behavior confirmed by the simulations. Indeed, increasing the pump power up to our 20 mW maximum does not result in a significant increase in DW power (see Fig. 4b). This could be due to the low values of $|\beta_2|$ in these waveguides, leading to a longer compression point than in the two larger waveguides (see Supplementary Note 1). Therefore, in narrower waveguides, spectral broadening caused by modulation instability[46] can happen on the same length scale of soliton compression, already for coupled pump powers in the 12–20 mW range. In fact, at a given point on the waveguide, modulation instability scales exponentially with the pump power[43]. Equivalently, one can look at the soliton number, which also increases when $|\beta_2|$ decreases. Indeed, it has been shown that soliton fission occurs earlier than wave breaking by modulation instability[35] when $N < 10$.

In Fig. 4b, we report numerical and experimental data of the mid-IR DW power as a function of the coupled pump power. In the employed pump power range, all waveguides can generate mid-IR DW with at least 4.5 mW but, due to the lower CE, the required pump power increases with the wavelength reach. However, thanks to the lower in-coupling losses, we could inject more power in the two larger waveguides and reach stronger DW generation at longer wavelengths. In fact, in such samples, the CE increases all over the tested power range, as $N > 10$ is expected for coupled power beyond 40 mW. Overall, this means that more than 1 mW can be estimated at the waveguides output (see Methods), covering the entire 3–4 μm spectral region.

We could also experimentally estimate the visible DW contribution by changing the output objective with a silica lens and the long pass filter with a short pass one with cut-off at 700 nm. While this configuration leads to larger uncertainties in the measurements, mainly due to larger scattering losses at shorter wavelengths, both simulations and experimental data confirmed the much lower CE for the visible DW generation, which remained well below 5%.

**Proof-of-principle spectroscopy measurement of $C_2H_2$.** To demonstrate the application of our on-chip source, we preformed mid-IR absorption spectroscopy of acetylene ($C_2H_2$) using the DW near 3 μm from the 1000 nm wide waveguide (see Methods for additional information on the employed experimental set-up). The light from the $Si_3N_4$ waveguide is directly coupled into a 108.5 cm long gas cell, which contained the sample or reference gas. The transmitted spectra measured with an optical spectrum analyzer (OSA) after the cell, with and without sample $C_2H_2$ gas, are shown in Fig. 5a. Absorption dips originating from $C_2H_2$ in the gas cell and from atmospheric gas ($H_2O$) outside the cell are clearly seen. Figure 5b, c shows the normalized absorption spectrum of $C_2H_2$ compared to the one simulated utilizing the HITRAN database[47]. The standard deviation of residual absorbance for this global fit is $\sim 2.5 \times 10^{-4}$ cm$^{-1}$. The maximum absorbance for 396 ppm $C_2H_2$ was measured to be $8.8 \times 10^{-3}$ cm$^{-1}$. The signal-to-noise ratio was calculated to be 35 and the noise equivalent concentration is $\sim 11$ ppm. Clear spikes in the residuals near the absorption peaks are likely due to the wavelength nonlinearity of the OSA within the detection range, which is not taken into account in the model.

## Discussion

We showed that the generation efficiency of mid-IR DW from a commercial femtosecond SWIR fiber laser can be greatly

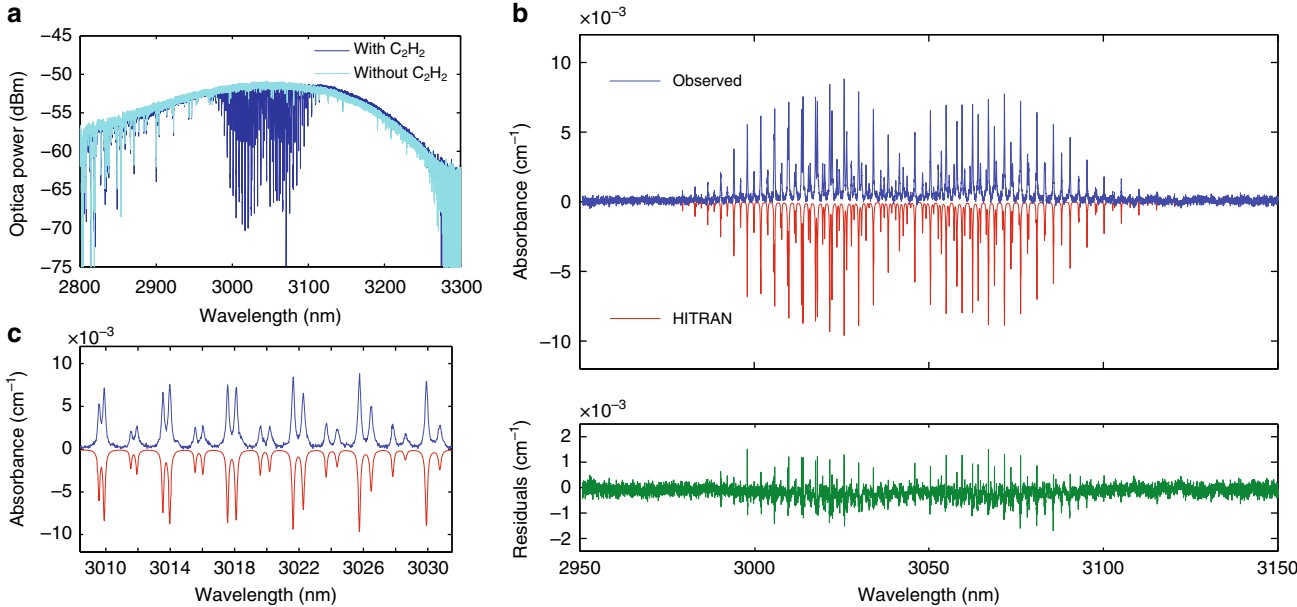

**Fig. 5** Mid infrared gas spectroscopy. **a** Transmitted spectrum of the empty (light blue) or filled (dark blue) gas cell with $C_2H_2$ in $N_2$ buffer gas at 1 atm total pressure. **b** Observed absorbance (dark blue) with the gas cell filled with $C_2H_2$ in $N_2$ buffer gas by normalizing the sample spectrum to that of the pure $N_2$ reference gas in the cell, theoretical absorbance (HITRAN) (in red and inverted for clarity), and the residuals (green), differences between simulations and experimental data, of the multiple-line fit. **c** Zoom on the observed and HITRAN absorbance

enhanced to more than 35%, by proper dispersion engineering of $Si_3N_4$ waveguides. In fact, state-of-the-art fabrication techniques and fiber laser technology allow for careful waveguide design which, in combination with a proper choice of the pump parameters, significantly favor the generation of mid-IR DW and reduce the visible one, allowing us to report the highest efficiency measured to date for on-chip mid-IR DW generation. The central wavelength of the DWs increases with the width of the waveguide and can be precisely positioned inside the first mid-IR transparency window while maintaining preferential power transfer to the mid-IR. Although the efficiency decreases with increasing DW wavelength, we were able to obtain, at the chip output, a maximum average output power of more than 1 mW spanning roughly the entire 3–4 μm region. These values represent a significant improvement both in terms of spectral coverage and efficiency, compared to previous mid-IR SCG[34]. Notably, the mid-IR output energy we obtained is comparable to the state-of-the-art mid-IR sources for dual-comb spectroscopy, based on DFG in PPLN waveguides[13], pumped with similar peak power fiber frequency combs. The source can be used for absorption spectroscopy, as shown here with the detection of $C_2H_2$ in the 3 μm band. The expected comb structure of the DW[34] could also enable dual-comb measurements, which however would be best implemented pumping the waveguides using a laser with hundreds of MHz of repetition rate[8].

In summary, the presented approach can lead to a very efficient, compact, and easy to use device for coherent mid-IR light generation. In fact, it benefits from photonic integration both for the chip-scale nonlinear stage, compatible with planar fabrication techniques, and for the pump source which is a silica fiber-based laser. The device can cover the 3–4 μm region, which hosts the signature of important greenhouse gases like methane ($CH_4$) and nitrous oxide ($N_2O$). The absorption lines of carbon dioxide ($CO_2$), just at the boundaries of the covered range (around 2.7 and 4.3 μm), can be targeted with slightly narrower or wider waveguides. In addition, this approach provides a power level sufficient for spectroscopy application[13], bridging the gap between fiber sources and quantum cascade lasers, which are the

workhorse of mid-IR spectroscopy devices. Such result could therefore provide a suitable alternative to microresonators[48–52] to generate mid-IR frequency combs on a chip, when sub-gigahertz teeth spacing is required. Moreover, soliton-induced SCG in integrated photonic platforms has been recently demonstrated to coherently broaden the spectrum of optical frequency combs by more than one octave[53], allowing their stabilization in f-to-2f schemes[54,55]. Therefore, with the possibility to combine both $\chi^{(2)}$ and $\chi^{(3)}$ nonlinearities in $Si_3N_4$[56–58], full on-chip stabilization of the mid-IR comb generated in these waveguides can also be considered.

## Methods

**Numerical simulations**. The waveguide dispersion was simulated using a Finite Element Method software (COMSOL Multiphysics) in which the wavelength dependence of the real part of the refractive index of $Si_3N_4$ and $SiO_2$ were included using the Sellmeier equation reported in refs. [48,59] respectively. In Fig. 1b, we include the imaginary part of the refractive index of $SiO_2$ as $k = 5 \cdot 10^{-(9+\lambda)}$, which qualitatively reproduces, in the region from 1 to 5 μm, the data reported in ref. [60]. The SCG was simulated based on the nonlinear Schrödinger equation. We consider an input $sech^2$ pulse with a frequency chirp of $-1000$ fs$^2/2\pi$, leading to pulse duration of about 110 fs. The laser temporal pulse broadening mainly comes from the dispersion in the wave-plates and input objective. We included the spectral dependence of the nonlinear coefficient and a full and complete dispersion profile that comprises up to 30th order when Taylor expanded with respect to the pumping frequency. Including more than 30 HOD terms does not change the simulation result but just slowed down the computation. We also set the Raman fraction and linear propagation losses to 0. In order to better compare with the experimental results, we included a nonlinear contribution in the experimental in-coupling losses proportional to the coupled pump power to the third power. We attribute the need of this correction to the increase of multiphoton absorption processes and the consequent free carrier absorption, mainly coming from the increase of visible DW and the generated third harmonic with the pump power.

**Waveguide fabrication**. The waveguides under test are fabricated accordingly to the photonic damascene process[36], which consists of a conformal low pressure chemical vapor deposition (LPCVD) of $Si_3N_4$. In addition to the waveguides, a dense filler pattern is patterned into the hard mask of amorphous silicon on a 4 μm thick wet thermal silicon dioxide. This pattern efficiently releases the tensile stress and prevents cracking of the thick $Si_3N_4$ thin film. The $Si_3N_4$ core channels are covered by a cladding of low temperature oxide (LTO).

**Dispersive wave generation efficiency**. We estimate the DW power by integration on the FT-OSA. To first calibrate the power measurement in the FT-OSA, we directly coupled the attenuated pump laser at 2090 nm to the MMF via the parabolic mirror and sent it to the spectrometer. We measured the value obtained integrating the entire laser bandwidth in the FT-OSA ($P_{FT-OSA}$) and we compared it to the power detected with an InGaAs photodiode (Thorlabs S148C) at the output of the MMF ($P_{PD}$). The quantity $c = P_{PD}/P_{FT-OSA}$ gave us the calibration factor for the spectrometer. The mid-IR DWs were then integrated over their entire spectral extend, namely 86–113, 80–110, 76–105, and 70–90 THz for the 1000, 1050, 1100, and 1175 nm wide waveguides, respectively. These values were then multiplied by the measured calibration factor ($c$) at the pump wavelength to retrieve the mid-IR DW power at the MMF output. It is important to notice that $c$ is constant over the entire FT-OSA spectral range. Finally, we considered the total out-coupling losses, including transmission through the MMF and the output lens, which was optimized for mid-IR throughput. The on-chip CE is then calculated as the ratio between the on-chip mid-IR DW power over the coupled pump power. The coupled pump power is estimated by direct detection of the pump laser before the chip, and taking into account the in-coupling losses from the input lens (5 dB).

Also, we simulated the CE by integrating the output spectra obtained solving the nonlinear Schrödinger equation. The CE was defined as the ratio between the integral performed over the DW bandwidth, over the integral of the input spectrum. The on-chip power was then calculated by multiplying the theoretical CE by the value of the pump power used in the simulation.

**Absorption spectroscopy**. The mid-IR DW is collimated by adjusting the focal distance of the objective at the chip output and passes through the cell. The sample spectrum is obtained by filling the gas cell with 396 ppm $C_2H_2$ buffered in $N_2$ in 1 atm total pressure at $T = 296$ K. The reference is obtained by purging the cell with pure $N_2$. The measurement time is ~2 min. The light exiting the gas cell is guided to an OSA (Yokogawa AQ6376) through a single mode indium fluoride (InF$_3$) fiber to improve spectral resolution. In this configuration, the total losses from the chip output are estimated to be around 20 dB. The OSA was set to its best resolution, corresponding to 0.1 nm with a single mode fiber, and high sensitivity setting using the internal chopper mode. We used 0.1 sampling interval for Fig. 5a and 0.02 for Fig. 5b, c.

## Data availability

The data that support the findings of this study are available from the corresponding authors on reasonable request.

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

## Acknowledgements

D.G., E.T., and C.-S.B. acknowledge support by the European Research Council (ERC) under grant agreement ERC-2012-StG 306630-MATISSE and ERC-2017-CoG 771647-PISSARRO. C.H., H.G., and T.J.K. acknowledge support by contract W31P4Q-16-1-0002 (SCOUT) from the Defense Advanced Research Projects Agency (DARPA), Defense Sciences Office (DSO), and support by the Swiss National Science Foundation under grant agreement no. 161573; H.G. acknowledges funding from the European Union's Horizon 2020 research and innovation program under Marie Sklodowska-Curie IF grant (No. 709249); F.Y. acknowledges support by the Swiss National Science Foundation under grant agreement no. 178895. The authors acknowledge Prof. Luc Thévenaz for fruitful discussions.

## Author contributions

D.G. performed the simulations, and D.G. and E.T. performed the experiments under the supervision of C.-S.B. H.G. and C.H. conceived the design of large cross-section $Si_3N_4$ waveguides under the supervision of T.J.K. C.H. fabricated the large cross-section waveguides. F.Y., E.T., and D.G. performed the gas absorption experiment and F.Y. performed the HITRAN simulation. All authors discussed the data. D.G. and E.T. wrote the manuscript with input from others. C.-S.B. supervised the project.

## Additional information

**Competing interests:** The authors declare no competing interests.

