## [Peer Review File · Nature Communications]

Reviewers' comments:

Reviewer #1 (Remarks to the Author):

The paper "Mid infrared gas spectroscopy using efficient fiber laser driven photonic chip-based supercontinuum", by Grassani et al., reports on the generation of supercontinuum (SC) in the mid-IR using Si₃N₄ waveguides pumped around 2 microns. The authors show that the judicious design of these waveguides to engineer the overall dispersion is viable to optimize the dispersive wave (DW) generation and the conversion efficiency within a selected spectral region (i.e. 3 – 4 μm). They further demonstrate the use of their system for a proof of principle demonstration of absorption spectroscopy in the spectral region around 4 μm, important for the detection of greenhouse gases (as C₂H₂ in their current work).

Overall, I found the paper interesting and overall well-written. I think that the proof-of-concept demonstration of spectroscopy is significant, while the overall approach and results are valid (i.e. spectral coverage and mW output power). Conversely, it appears to me that the overall conceptual novelty (dispersion engineering + DW control) is somehow limited, and mostly incremental to the use of a particular target spectral window (i.e. 3 – 4 μm) and source wavelength (i.e. 2 μm fs-fiber laser – in particular with respect to their previous work Ref. 29). The integrated aspect is of course significant, but the claimed record 35% power conversion efficiency (CE) has to be put in perspective when considering that this figure is based on an estimated coupled on-chip power after more than 11 dB of coupling losses. That being said, I think that the manuscript is interesting and should warrant publication in Nature Communications, after their work has been put in a better context and the points below have been addressed.

-- General comments:

1/ As this constitute one the main claim for the paper, giving several comparative numbers in terms of output power, conversion efficiency (total and coupled) or bandwidth/spectral density compared to other methods for Mid-IR SC generation (fiber/integrated platforms for various/ most common materials) would be useful for the reader and constitute a better baseline to assess the promises of this work. A few key numbers in the main text, and ideally a (succinct) supplementary table would be really appreciated.

2/ The bibliography could maybe gain in relevance by addressing the concepts of multiple zero-dispersion wavelengths supercontinuum generation as well as dispersive wave generation, trapping and controlled steering (via e.g. tapered fibres, event-horizons analogue, accelerating pulses) in a more general manner, as these topics were intensely studied over the years in a fibered context at telecom wavelengths, here essentially transposed to mid-IR. I also think that key papers in mid-IR and/or integrated SC generation from other groups could also be benefit in providing a better context to the paper. (e.g. <https://doi.org/10.1364/OE.20.001685> ; <https://doi.org/10.1364/OE.18.000923> ; <https://doi.org/10.1364/OL.41.001756>)

-- Specific comments:

Page 4: "leading to an initial spectral broadening caused by self-phase modulation (SPM) and subsequent temporal compression which are proportional to the soliton number" => Maybe it would be worth mentioning/introducing the concept of modulation instability and recurrence here, especially when mentioning later on the concept of maximal compression point ?

Figure 1: It would be useful and clearer to see all 3 panels use the same wavelength span (with eventual insets wherever relevant)

Page 6: "commercial, turnkey soliton self-frequency shifted thulium-doped fiber mode locked laser (NOVAE Brevity λ+,) with pulse duration at full width half maximum (FWHM) of 78 fs and

repetition rate of 19 MHz" => I would mention the initial laser average power and bandwidth, and add the input spectrum on Fig. 2b (or as an inset)

Figure 2: An axis scale in the power spectral density of Fig. 2b is required.

Page 6-7: "Indeed the obtained simulated spectrum reproduces well the experimental one" => A description of the simulation parameters, NLSE and terms used (higher order term and value used in the dispersion Taylor expansion, self-steepening, losses, reasons to set the Raman fraction to zero) would be appreciated in the Methods section. Some discrepancies are also seen in the central part close to the pump and in the shape of the spectrum. Can the authors comment on the possible reasons for such discrepancies? Raman effects neglected ?

Page 7: "which can be primarily due to the recoil of the central soliton toward shorter wavelengths" => I think that this concept should be explained briefly in the text (eventually more in the Supplementary) and accompanied with a reference for the more specialist reader. Can such dynamics also be attributed to other higher-order effects that are neglected in the NLSE model ?

Page 7: "We estimate the DW power by integration on the FT-OSA, which was calibrated against a known signal and known photodiode." => I think that this experimental approach should be explained succinctly in the Method, and at the very least, the spectral boundaries used for integration should be given in the text for the reader.

Page 8: "Additionally, the on-chip CE for all tested waveguides was simulated" => How was it simulated ? Using the NLSE approach used to retrieve the spectrum in Fig. 2b ? This should be mentioned and explained in the Method (bandwidth of power extraction/maximal spectral density etc.)

Page 8-9: "Indeed increasing the pump power, up to our 20 mW maximum, does not result in a significant increase in DW power (see Fig. 4b). This could be due to the low values of 2β in these waveguides, leading to an already large soliton number (> 10) for coupled pump power in the 12-20 mW. This plateau is not observed in the two largest waveguides the soliton number remaining well below 10 for all tested coupled powers, and soliton-related dynamics being clearer." => It would be useful to see this quantitative conversion efficiency from different aspect. Here, there is uniquely a consideration in term of power over a given bandwidth. How does this plateau behaves increasing/decreasing the power integration bandwidth ? The soliton number clearly impact the peak power, compression point, bandwidth and breathing period(s) of the subsequent soliton fission in perturbed high-order soliton dynamics. There is probably more to dig in this respect (power and periodicity of the compression vs conversion efficiency), but the explanation is here a bit too qualitative in my opinion. Additionally, such a plateau seems to disappear when considering the simulation results, but are only limited by the available coupled power in the waveguides. Also, why are the authors defining $N=10$ as a threshold for this behaviour ? It seems rather arbitrary but is there another relevant element not disclosed ?

Page 8-9: "This plateau is not observed in the two largest waveguides the soliton number remaining well below 10 for all tested coupled powers, and soliton-related dynamics being clearer" => This sentence is not clear in my opinion and should be reformulated. What are clearer soliton-related dynamics ??

Methods – Numerical Simulations: I would recommend succinctly detailing the model (GNLSE ?) and provide the variables used (dispersion, nonlinearity, self-steepening ? Raman ? etc.). Is 110 fs the chirped pulse duration ? Maybe giving the bandwidth or transform-limited duration would help the reader.

Methods – Absorption spectroscopy: "We used 0.1 and 0.02 sampling interval for Fig. 4a and Fig. 4b

respectively." => I do not understand if Figs 4 a and b represent the same data or refer to different measurement. How are these data expected to be different?

Supplementary: A color scale on the density data plot would be appreciated.

The references contain several typos:
[30] Second author is John Roy Taylor
[45] Lowercase "A"
[46] "Si3N4"
[47] Journal name is missing

Reviewer #2 (Remarks to the Author):

The authors demonstrate supercontinuum generation in a silicon nitride waveguide. The main novelty here is that they generate an asymmetric, powerful dispersive wave rather far from the pump through dispersive wave generation. This system is pumped by a fiber based system which makes the whole setup relatively simple.

In my opinion the technical quality of the paper is really high. The measurements are done in a reproducible manner, the supercontinuum is measured as function of different parameters: the power, the waveguide width,... However, the main issue I have is that the novelty is rather limited. I leave it open to the editor to make a decision to evaluate this, but I would like to make sure that the editor takes the following points in account:

-The waveguides discussed here on Figure 1 are the same (at least the figures are exactly the same) as in the Nat Phot paper published by the authors in 2018. Although the pump wavelength is different the authors do a very similar experiment there. In fact the supplementary information of that paper includes some simulations indicating that pumping such a large cross section waveguide at a longer waveguide is interesting since the dispersive wave shifts further into the mid-infrared.

-The idea to have a dispersive wave to extend a supercontinuum is not new, it has been shown on many integrated platforms and also in photonic crystal fibers and chalcogenide fibers. These are typically asymmetric because the group velocity dispersion of these fibers is very asymmetric around the pump.

-In fact, it would make more sense to pump a (chalcogenide, InF,..) fiber with the source than an integrated waveguide. This would make sense in terms of coupling losses handling,.. Such a product (2 um laser pumping InF) is commercial (https://www.thorlabs.com/newgrouppage9.cfm?objectgroup_id=10819) and has an efficiency of ~ 35%. A question arising is what can be added to this approach with an integrated solution?

Some additional short questions:

-The authors argue that they reach the greenhouse gas spectral region. CO2 absorbs at 4.3 um (not/barely reached in the paper) and the gas that is used in the gas cell (C2H2) is after a quick search not really considered an important greenhouse gas. I think the authors should remove the reference to the greenhouse gasses or make it clear what is meant here.

-From the paper it is not entirely clear how the conversion efficiency is calculated and over which band is integrated.

-Is the coupling efficiency over the whole band flat. I would expect that it is very wavelength dependent.

Reviewer #3 (Remarks to the Author):

The paper claims that they have dispersion engineered SiN waveguides to achieve record high conversion efficiencies in generating light in the 3-4 micron region. This is a very important area for spectroscopy and the reported efficiencies are impressive.

In general I would recommend publication, but have a few comments to address below.

The authors mention high yield, it would be interesting to know the actual yield numbers.

Page 9 2nd paragraph: It is unclear in this paragraph if 1 mW average power was actually experimentally measured at the output of the chip or if the authors are simply pointing out that it could be achieved.

In Figure 5c, the zoom shows the observed absorption vs the HITRAN absorbance. I would expect the measured absorption to have symmetric peaks similar to the HITRAN data, but they are not. The authors should provide an explanation for this and any impact it has on the validity of their measurement.

In the Discussion section the authors claim that they reach output power levels at 4 μ m which are comparable to the state-of-the-art mid-IR sources for dual comb spectroscopy, and that they are pumped with a similar fiber frequency comb. However, the repetition rate of the comb used in this paper is only 19 MHz, and the paper from (12) is at 200 MHz. They go on to say that they can obtain similar output power levels with roughly ten times less average pump power than (12). I believe this comment to be a bit misleading, as they are using a source that is roughly 10x less in repetition rate. The nonlinearity goes by peak power in the pulse, so if the authors switched to a 200 MHz source (in order to be similar with ref 12, they would require 10x more average power to achieve the same level of broadening. The comparison should be better explained.

The authors claim that their source could be used for dual-comb spectroscopy, but a 19 MHz source would not be useful for this as the comb modes are spaced too close together limiting the total optical bandwidth that could be sampled and the power per comb mode would be very low.

I suggest revising the claims made in the Discussion section per the comments above in order to make a more realistic comparison.

Point-by-point response to referees

Reviewers' comments:

Reviewer #1 (Remarks to the Author):

The paper “Mid infrared gas spectroscopy using efficient fiber laser driven photonic chip-based supercontinuum”, by Grassani et al., reports on the generation of supercontinuum (SC) in the mid-IR using Si₃N₄ waveguides pumped around 2 microns. The authors show that the judicious design of these waveguides to engineer the overall dispersion is viable to optimize the dispersive wave (DW) generation and the conversion efficiency within a selected spectral region (i.e. 3 – 4 μm). They further demonstrate the use of their system for a proof of principle demonstration of absorption spectroscopy in the spectral region around 4 μm, important for the detection of greenhouse gases (as C₂H₂ in their current work).

Overall, I found the paper interesting and overall well-written. I think that the proof-of-concept demonstration of spectroscopy is significant, while the overall approach and results are valid (i.e. spectral coverage and mW output power). Conversely, it appears to me that the overall conceptual novelty (dispersion engineering + DW control) is somehow limited, and mostly incremental to the use of a particular target spectral window (i.e. 3 – 4 μm) and source wavelength (i.e. 2 μm fs-fiber laser – in particular with respect to their previous work Ref. 29). The integrated aspect is of course significant, but the claimed record 35% power conversion efficiency (CE) has to be put in perspective when considering that this figure is based on an estimated coupled on-chip power after more than 11 dB of coupling losses. That being said, I think that the manuscript is interesting and should warrant publication in Nature Communications, after their work has been put in a better context and the points below have been addressed.

We thank the Reviewers for having read our manuscript and acknowledging the importance of this work warrant of publication in Nature Communications. In the following, together with a detailed answer to reviewer’s questions, we indicate what we added in the manuscript to better set our result in the general context of Mid-IR supercontinuum generation pumped by fiber lasers. We think now the strong points of our result are clearer.

-- General comments:

1/ As this constitute one the main claim for the paper, giving several comparative numbers in terms of output power, conversion efficiency (total and coupled) or bandwidth/spectral density compared to other methods for Mid-IR SC generation (fiber/integrated platforms for various/ most common materials) would be useful for the reader and constitute a better baseline to assess the promises of this work. A few key numbers in the main text, and ideally a (succinct) supplementary table would be really appreciated.

We thank the reviewer for the suggestion, and we agree that a better comparison would be useful for the reader. Generally, most of the Mid-IR Supercontinuum Generation (SCG) experiments using Mid-IR transparent nonlinear platforms were pumped by optical parametric oscillators with wavelength emission already in the Mid-IR, beyond 2.5 micron. These cases are technologically and also conceptually different to our work. In fact, in such configurations, it is obviously not possible to define a wavelength conversion efficiency from the short-wave infrared to mid-IR. In fact, the targeted region can be often achieved without the need of DWG. However, there are few demonstrations, both fiber- and chip- based, employing pump sources in the telecom or SWIR range. They include different mechanisms: SCG and single pumped Difference Frequency Generation (DFG). Moreover, some information like Mid-IR power and bandwidth or efficiency and coherence are often missing. Given the variety of methods and incomplete information at our disposal, we cannot provide a succinct table, but as suggested we extended the introduction including key performance parameters, when provided.

Herein we provide a detailed analysis:

An all fibered set-up can be found in [1], where a dispersion engineered InF₃ fiber is pumped with a mode-locked Thulium doped fiber laser at 1969 nm and generates a SC extending to 4.6 μm. The authors report conversion

efficiency to beyond 2200 nm of about 35%. This is a value comparable to our measurement, but the pulse energy is about 3 times the one used in our work (corresponding to 28 kW peak) and the long wavelength broadening is not clearly due to DWG but most probably to Raman. Also, the authors didn't measure the coherence of the Mid-IR part, but simulations suggest degradation with increasing fiber length, which in turns leads to a trade-off between coherence and extension of the Mid-IR spectrum. Large spectral broadening extending beyond 4.5 μm using a femtosecond telecom-band pump has been also demonstrated using Telluride Glass based fiber [2]. However, the lack of a complete GVD curve and theoretical simulations avoid further speculations on the nature of the Mid-IR portion of the SCG spectrum and conversion efficiency. In any case, given the high estimated soliton number, it is unlikely the SC can preserve the pump coherence in the Mid-IR part.

In [3] DWG is generated in a ChG nanospike with a very low pulse energy (about 18 pJ) utilizing a pump wavelength around 2000 nm. Moreover, the coherence of the Mid-IR part was demonstrated in a similar platform [4]. However, quantitative comparison is impossible as the authors didn't provide an estimate of the power generated into the Mid-IR DW. In general, Chalcogenide glasses are highly nonlinear platform and could provide very high conversion efficiency, especially when the core size is greatly reduced, like in this case. However, we think there are still some technological constraints which avoid power scaling to mW level Mid-IR radiation, despite the good efficiency.

- The large anomalous bulk dispersion needs very tiny structures to blue shift the ZDWL point. This often results in poor mode confinement of the fundamental mode into the core, and a non-negligible overlap with the silica cladding, needed to protect the fiber. In this case, the authors report losses of several dB/mm beyond 4 μm , therefore more than 10 times larger than in our case.
- The damage threshold, which happens for about 28 pJ in this case, limiting robust power scalability of the system.

As illustrated in the introduction of this manuscript, in terms of *photonic integrated waveguides*, it is very challenging to achieve the Mid-IR in the molecular fingerprint region beyond 3000 nm using a fiber laser. For instance, silicon waveguides didn't succeed yet to be correctly engineered to achieve this goal and Mid-IR DWG was achieved by pumping the waveguide with an OPO set near 2.5 μm [5]. In SiN, an interesting multicladding waveguide structure has been proposed in [6]. However, the detection range stopped at 2400 nm, avoiding the assessment of the spectral portion of interest for comparison with our work. Mid-IR extended SCG has been demonstrated in AlN waveguides [7]. The spectrum spans from 500 nm to 4000 nm when pumped with a telecom band femtosecond mode-locked fiber laser, coupling about 0,8 nJ (similar to our case). In this case the power generated inside the waveguide in the 3000 nm – 4000 nm range is about 0.3 mW, corresponding to a conversion efficiency of about 0.5%, similar to what we reported in our previous work using SiN (*Nat. Photonics* **12**, 330–335 (2018)). However, it's not demonstrated if AlN waveguides can be correctly dispersion engineered and have low enough Mid-IR losses to compare with our current efficiency when pumping in the two micron region.

We would like to stress that, despite the clear spectroscopic oriented motivation of the above mentioned SCG works; none of them reported a spectroscopy measurement with the generated Mid-IR radiation. Usually, Mid-IR spectroscopy is carried out from near-IR fiber lasers exploiting DFG in $\chi^{(2)}$ crystals. Recently, there has been a trend in miniaturizing and simplifying DFG schemes by utilizing single pump configuration with chip-scale nonlinear platforms, like for SCG. It is thus interesting to expand our comparison with intrapulse based DFG.

DFG using a *single* fiber femtosecond source was also reported in the same AlN waveguide used in [7]. Here DFG generates radiation up-to 5.5 μm involving the pump and the long-wavelength dispersive wave. However, from the reported spectra, DFG seems to be lower than the above-mentioned Mid-IR DWG one.

In [8] intrapulse DFG, generated from an Er:doped fiber MLL, enables tunable Mid-IR radiation in the 4-5 μm region using PPLN waveguides. A cascaded $\chi^{(2)}$ process allows the pump pulse spectral broadening for intrapulse DFG. Few cm long PPLN waveguides led to conversion efficiency in the Mid-IR of about 0.1 % over a quite narrow bandwidth (about 20 nm). Using a-periodically poled LN waveguides up to 150 nm bandwidth has been reported by the authors, who claimed power scaling up to mW level in the Mid-IR with similar efficiency to the case of PPLN waveguides.

References

1. R. Salem, Z. Jiang, D. Liu, R. Pafchek, D. Gardner, P. Foy, M. Saad, D. Jenkins, A. Cable, and P. Fendel, "Mid-infrared supercontinuum generation spanning 1.8 octaves using step-index indium fluoride fiber pumped by a femtosecond fiber laser near 2 μm ," *Opt. Express* **23**, 30592 (2015).
2. P. Domachuk, N. A. Wolchover, M. Cronin-Golomb, A. Wang, A. K. George, C. M. B. Cordeiro, J. C. Knight, and F. G. Omenetto, "Over 4000 nm bandwidth of mid-IR supercontinuum generation in sub-centimeter segments of highly nonlinear tellurite PCFs," *Opt. Express* **16**, 7161 (2008).
3. N. Granzow, M. A. Schmidt, W. Chang, L. Wang, Q. Coulombier, J. Troles, P. Toupin, I. Hartl, K. F. Lee, M. E. Fermann, L. Wondraczek, and P. S. J. Russell, "Mid-infrared supercontinuum generation in As₂S₃-silica "nano-spike" step-index waveguide," *Opt. Express* **21**, 10969 (2013).
4. K. F. Lee, N. Granzow, M. A. Schmidt, W. Chang, L. Wang, Q. Coulombier, J. Troles, N. Leindecker, K. L. Vodopyanov, P. G. Schunemann, M. E. Fermann, P. S. J. Russell, and I. Hartl, "Midinfrared frequency combs from coherent supercontinuum in chalcogenide and optical parametric oscillation," *Opt. Lett.* **39**, 2056–2059 (2014).
5. R. K. W. Lau, M. R. E. Lamont, A. G. Griffith, Y. Okawachi, M. Lipson, and A. L. Gaeta, "Octave-spanning mid-infrared supercontinuum generation in silicon nanowaveguides.," *Opt. Lett.* **39**, 4518–21 (2014).
6. J. M. Chávez Boggio, A. Ortega Moñux, D. Modotto, T. Fremberg, D. Bodenmüller, D. Giannone, M. M. Roth, T. Hansson, S. Wabnitz, E. Silvestre, and L. Zimmermann, "Dispersion-optimized multicladding silicon nitride waveguides for nonlinear frequency generation from ultraviolet to mid-infrared," *J. Opt. Soc. Am. B* **33**, 2402 (2016).
7. D. D. Hickstein, H. Jung, D. R. Carlson, A. Lind, I. Coddington, K. Srinivasan, G. G. Ycas, D. C. Cole, A. Kowligy, C. Fredrick, S. Droste, E. S. Lamb, N. R. Newbury, H. X. Tang, S. A. Diddams, and S. B. Papp, "Ultrabroadband Supercontinuum Generation and Frequency-Comb Stabilization Using On-Chip Waveguides with Both Cubic and Quadratic Nonlinearities," *Phys. Rev. Appl.* **8**, 1–8 (2017).
8. A. S. Kowligy, A. Lind, D. D. Hickstein, D. R. Carlson, H. Timmers, N. Nader, F. C. Cruz, G. Ycas, S. B. Papp, and S. A. Diddams, "Mid-infrared frequency comb generation via cascaded quadratic nonlinearities in quasi-phase-matched waveguides," *Opt. Lett.* **43**, 1678–1681 (2018).

Action taken

We changed the introduction including the above arguments and references. They can be found at page 3 of the revised version, in the following way:

"Broadband mid-IR generation can also be achieved by supercontinuum generation (SCG) in soft glass optical fibers^{14–23} or photonic integrated waveguides^{24–31}. Compared to other wavelength conversion schemes, SCG offers some advantages³²: it is in fact a compact single pass geometry that does not require any additional seed laser or temporal synchronization, and it can provide a broader and tunable mid-IR emission. When CMOS compatible materials are employed, SCG platforms benefit from lithographic precision and high yield, and usually have low power consumption. Recent works have shown that such approach can be applied as well to dual-comb spectroscopy³².

Often, to extend the reach into the mid-IR, the nonlinear waveguides are pumped with an optical parametric oscillator (OPO) placed beyond the 2 micron wavelength range, but there is interest in driving such platforms with femtosecond mode-locked fiber lasers, which are reliable, easy to use and compact frequency comb sources.

However, up to now, very few demonstrations succeeded in generating supercontinua following this paradigm. High energy (2-3 nJ) femtosecond pulses in the 2 micron region can lead to more than 30% of conversion efficiency (CE) beyond 2200 nm in a InF₃ fiber²², and spectral broadening extending beyond 4.5 μm in Telluride photonic crystal fibers²³. Despite the large CE and spectral extent, the coherence of such mid-IR emission has not been

experimentally investigated. Conversely, low pulse energy (18 pJ) at 2000 nm can lead to coherent mid-IR SCG in ChG nanospikes^{16,17} or nanotapers¹⁹, but early damage power threshold and high mid-IR absorption from the cladding avoid power scaling up to the mW level. In terms of chip integrated waveguides, mid-IR extended SCG has been experimentally demonstrated in AlN waveguides pumped with 0.8 nJ from a telecom band femtosecond mode-locked fiber laser²⁴. The power generated inside the waveguide in the 3000 nm – 4000 nm range is about 0.3 mW, corresponding to a CE close to 0.5 %. Overall, none of the above-mentioned approaches has been employed for mid-IR spectroscopy demonstrations.

Also, in the last years there has been a trend in miniaturizing and simplifying DFG schemes by utilizing single pump configuration with chip-scale nonlinear platforms. Emission up to 5.5 μm was obtained by DFG between the pump and the long-wavelength dispersive wave (DW) in AlN waveguides²⁴, while intrapulse DFG, generated from an Er:doped fiber mode-locked laser, enables tunable mid-IR radiation in the 4-5 μm region using PPLN waveguides³³. However, efficiencies are limited to below 0.5 %.

Recently, we showed that direct generation of mid-IR light, from an erbium-doped fiber laser at 1.56 μm , is possible through DW generated in Si_3N_4 , and asserted the phase coherence and a frequency comb nature³⁴. This platform has the potential to merge all the desired features addressed separately in the above-mentioned devices: a power scalable, fiber laser pumped coherent mid-IR generation in a low loss chip-scale waveguide with lithographical control of its dispersion.”

2/ The bibliography could maybe gain in relevance by addressing the concepts of multiple zero-dispersion wavelengths supercontinuum generation as well as dispersive wave generation, trapping and controlled steering (via e.g. tapered fibres, event-horizons analogue, accelerating pulses) in a more general manner, as these topics were intensely studied over the years in a fibered context at telecom wavelengths, here essentially transposed to mid-IR. I also think that key papers in mid-IR and/or integrated SC generation from other groups could also be benefit in providing a better context to the paper. (e.g. <https://doi.org/10.1364/OE.20.001685> ; <https://doi.org/10.1364/OE.18.000923> ; <https://doi.org/10.1364/OL.41.001756>)

We thank the referee for pointing out the additional references from the integrated nonlinear optics community. The concept of multiple ZDW points was indeed already treated in the section “Design of mid-IR source through dispersive wave optimization” referring to *Opt. Lett.* 34, 2072 (2009) and *Phys. Rev. A* 79,1-6(2009). For completeness, we included a comparison with similar nonlinear mechanism reported in fiber optics and we expanded the bibliography accordingly.

Action taken

We added the first work suggested by the Reviewer: *Opt. Express* 20, 1685 (2012), now reference 42, where the authors reported numerical simulation of silicon waveguides exhibiting 4 ZDWL points to flatten the SCG. The third work suggested by the referee (now reference 21) has been added in the introduction: “*Broadband mid-IR generation can also be achieved by supercontinuum generation (SCG) in soft glass optical fibers¹⁴⁻²³*”. As for the second work suggested, we decided to don’t include it. It is based on SCG in a silica waveguide, which is not suitable platform in the Mid-IR. Moreover, the SCG dynamics is different to our case and cannot be considered as a suitable reference in the sub-section “Design of mid-IR source through dispersive wave optimization”. In fact, although the dispersion profile has two ZDW points like in our case, the pump is placed very close to them and not entirely in the anomalous dispersion region. This leads to SCG dynamic which is different from the standard soliton one, reported in our work. Instead, we added reference 38 *Phys. Rev. A* 51, 2602–2607 (1995) for an analytical treatment of DWs generated by high order dispersion terms.

With respect to DWG and soliton dynamics observed over the past years in fibers, in our work there is the fundamental difference of the weak Raman response of SiN (we added the work reported in reference 39 to support this claim). This ultimately reduces the soliton red shift and consequent pulse slow down. In our case, the main responsible for pulse shifting (spectrally and temporally) are self-steepening and recoil. We include this concept at the bottom of page 8, referring for comparison with fibers, to Gorbach, A. V. & Skryabin, D. V. *Nat. Photonics* 1, 653–657 (2007) and Wang *Opt. Express* 21, 11215 (2013), in the following way:

“In fact, because of the negligible Raman response³⁹, we don’t see the red-shift of the soliton and the consequent trapping of DW, as observed in silica fibers^{44,45}.”

Also, in the Supplementary Material, we better specify the soliton dynamics:

“The wavelength shift due to the soliton recoil can be observed in the spectra reported in Supplementary Fig.1 (e)-(h) too. Also, this effect can be noticed in the temporal domain. In fact, after a first slowdown of the soliton pulse due to an increase of its group index due to self-steepening, the pulse accelerates reducing its delay in the proximity of the compression point. This is a signature of recoil, as a spectral blue-shift accelerates a pulse placed in the anomalous dispersion region. Consistently with the above-mentioned arguments, this bend in the pulse trajectory is more pronounced in the smaller waveguides (see Supplementary Fig. (i)-(n)).”

-- Specific comments:

Page 4: “leading to an initial spectral broadening caused by self-phase modulation (SPM) and subsequent temporal compression which are proportional to the soliton number” => Maybe it would be worth mentioning/introducing the concept of modulation instability and recurrence here, especially when mentioning later on the concept of maximal compression point ?

At page 4, we describe the design rules to follow in order to efficiently generate a dispersive wave in the mid-IR, when pumping the waveguide in the femtosecond pulse regime. In this regime, high order soliton dynamics triggers soliton compression earlier than modulation instability (MI), which is thus not mentioned. However, it is true that increasing the pump power, MI can compete with soliton compression on a similar length scale. In fact, in a given point along the waveguide, MI scales exponentially with the coupled pump power. Indeed, this is at the origin of the plateau in the on-chip Mid-IR power observed in Fig.4b.

Action taken

At page 10 of the new version, in the sub-section “Evolution and efficiency of DW generation”, we now explicitly mention MI. Also, we included a reference to (*Opt. Express* **13**, 3989 (2005)) which studies the wave breaking caused by MI including the effect of GVD:

“Indeed, increasing the pump power up to our 20 mW maximum does not result in a significant increase in DW power (see Fig. 4b). This could be due to the low values of $|\beta_2|$ in these waveguides, leading to a longer compression point than in the two larger waveguides (see Supplementary Figure 1). Therefore, in narrower waveguides, spectral broadening caused by modulation instability⁴⁶ can happen on the same length scale of soliton compression already for coupled pump powers in the 12-20 mW range. In fact, at a given point on the waveguide, modulation instability scales exponentially with the pump power⁴³.”

Figure 1: It would be useful and clearer to see all 3 panels use the same wavelength span (with eventual insets wherever relevant)

In this specific case, we would prefer to keep the axis as they are. In fact, extending the wavelength range of Fig. 1a up to 5 μm needs an extension of the y-axis too, as the GVD will largely increase beyond 3 micron. This would vertically compress the curves around zero making almost indistinguishable the shift in the second ZDW, which is the feature we want to stress here, rather than the GVD in the molecular fingerprint region. Analogously, Fig1b and c needs to expand up to 5 micron, to clearly show the Mid-IR capabilities of large cross section waveguides in terms of confinement loss and phase-matching.

Action taken

Please note that we changed Fig1b. In the previous version, the waveguide losses were computed considering a constant imaginary part of the silica cladding (k) in the 3-5 micron region. This overestimates the losses below 4 micron and underestimates them beyond. We corrected this picture by including a wavelength dependence of k in our simulation, together with the absorption coefficient expressed in dB/mm, a quantity more familiar to the readers.

We added a paragraph to the Methods section to better explain this:

“The waveguide dispersion was simulated using a Finite Element Method Software (COMSOL Multiphysics) in which the wavelength dependence of the real part of the refractive index of Si_3N_4 and SiO_2 were included using the Sellmeier equation reported in⁴⁸ and⁵⁸ respectively. In Fig.1b, we include the imaginary part of the refractive index of SiO_2 as $k = 5 \cdot 10^{-(9+\lambda)}$, which qualitatively reproduces, in the $1 - 5 \mu\text{m}$, the data reported in⁵⁹.”

Page 6: “commercial, turnkey soliton self-frequency shifted thulium-doped fiber mode locked laser (NOVAE Brevity $\lambda+$,) with pulse duration at full width half maximum (FWHM) of 78 fs and repetition rate of 19 MHz” => I would mention the initial laser average power and bandwidth, and add the input spectrum on Fig. 2b (or as an inset)

We thank the referee for this suggestion.

Action taken

We included this information in the text (beginning of page 7):

“The pump source is a commercial, turnkey soliton self-frequency shifted thulium-doped fiber mode locked laser (NOVAE Brevity $\lambda+$,) with pulse duration at full width half maximum (FWHM) of 78 fs, bandwidth of 60 nm centered at 2090 nm, repetition rate of 19 MHz and average power of about 100 mW.”

Also, we included in Fig.2b the simulated and measured laser input spectra.

Figure 2: An axis scale in the power spectral density of Fig. 2b is required.

We updated Fig. 2b with this information.

Page 6-7: “Indeed the obtained simulated spectrum reproduces well the experimental one” => A description of the simulation parameters, NLSE and terms used (higher order term and value used in the dispersion Taylor expansion, self-steepening, losses, reasons to set the Raman fraction to zero) would be appreciated in the Methods section. Some discrepancies are also seen in the central part close to the pump and in the shape of the spectrum. Can the authors comment on the possible reasons for such discrepancies? Raman effects neglected ?

Please notice that the β_2 values at the pump wavelength were already reported in the text for all the investigated waveguides.

The Raman shock term in SiN has been measured to be much lower with respect to silica (PRL 116, 103902 (2016)). Indeed Intrapulse Raman Scattering and consequent soliton frequency shift has never been reported in SiN SCG.

The discrepancies observed in the central part of the pump spectrum cannot be solved by including Raman terms in the simulations. We think they are most likely due to the fact that we optimized the focal length of the output lens to have the best throughput in the mid-IR DW region. This could result in a lower transmission at shorter wavelengths.

Action taken

We included the following details in the updated Methods section:

“We included the spectral dependence of the nonlinear coefficient and a full and complete dispersion profile that comprises up to 30th order when Taylor expanded with respect to the pumping frequency. Including more than 30 HOD terms didn't change the simulation result and just slowed down the computation.”

Page 7: “which can be primarily due to the recoil of the central soliton toward shorter wavelengths” => I think that this concept should be explained briefly in the text (eventually more in the Supplementary) and accompanied with a reference for the more specialist reader. Can such dynamics also be attributed to other higher-order effects that are neglected in the NLSE model ?

In our model, we include self-steepening and complete dispersion profile that comprises up to 30th order when Taylor expanded with respect to the pumping frequency. We set the Raman fraction to zero, but including the Raman fraction would indeed lead to the opposite effect: a blue-shift of the Mid-IR DW, as a consequence of the red-shift of the central soliton by Intrapulse Raman Scattering. Indeed, in the Supplementary material, we show that considering numerical simulations performed with our NLSE model, the wavelength emission of the Mid-IR DWG is correctly predicted.

Action taken

To be clearer, we now included a sentence (underlined in the following) in page one of the Supplementary Materials:

“Supplementary Fig.1 (a)-(d) clearly show that the discrepancy in the position of DW generation observed between Fig.3a and b in the main text mainly come from the larger blue shift of the central soliton due to the stronger recoil, not included in the phase matching condition of Equation 1 in the main text, underwent in smaller waveguides, and given by the larger soliton number and efficiency in power transfer to the DW. In fact, comparing the experimental data with the spectrum obtained by numerically solving the Nonlinear Schrödinger Equation (NLSE), we see how the predicted DW well reproduces the position of the experimental one for all the waveguides”

As for better show the spectral re-coil contribution, we added in the same section:

“Also, this effect can be noticed in the temporal domain. In fact, after a first slowdown of the soliton pulse due to an increase of its group index due to self-steepening, the pulse accelerates reducing its delay in the proximity of the compression point. This is a signature of recoil, as a spectral blue-shift accelerates a pulse placed in the anomalous dispersion region. Consistently with the above mentioned arguments, this bend in the pulse trajectory is more pronounced in the smaller waveguides (see Supplementary Fig. (i)-(n)).”

Page 7: “We estimate the DW power by integration on the FT-OSA, which was calibrated against a known signal and known photodiode.” => I think that this experimental approach should be explained succinctly in the Method, and at the very least, the spectral boundaries used for integration should be given in the text for the reader.

We thank the reviewer for pointing out the lack of completeness of this part of our manuscript.

Action taken

We added the section “**Dispersive wave generation efficiency**” to the Methods where we included more details on the experimental measurement of the CE:

“We estimate the DW power by integration on the FT-OSA. To first calibrate the power measurement in the FT-OSA, we directly coupled the attenuated pump laser at 2090 nm to the MMF via the parabolic mirror and sent it to the spectrometer. We measured the value obtained integrating the entire laser bandwidth in the FT-OSA (P_{FT-OSA}) and we compared it to the power detected with an InGaAs photodiode (Thorlabs S148C) at the output of the MMF (P_{PD}). The quantity $c = P_{PD}/P_{FT-OSA}$ gave us the calibration factor for the spectrometer. The mid-IR DWs were then integrated over their entire spectral extend, namely 86-113 THz, 80-110 THz, 76-105 THz and 70-90 THz for the 1000 nm, 1050 nm, 1100 nm and 1175 nm wide waveguides, respectively. These values were then multiplied by the measured calibration factor (c) at the pump wavelength to retrieve the mid-IR DW power at the MMF output. It is important to notice that c is constant over the entire FT-OSA spectral range. Finally, we considered the total out-coupling losses, including transmission through the MMF and the output lens, which was optimized for mid-IR throughput. The on-chip CE is then calculated as the ratio between the on-chip mid-IR DW power over the coupled pump power. The coupled pump power is estimated by direct detection of the pump laser before the chip, and taking into account the in-coupling losses from the input lens (5 dB)”.

Page 8: “Additionally, the on-chip CE for all tested waveguides was simulated” => How was is simulated ? Using the NLSE approach used to retrieve the spectrum in Fig. 2b ? This should be mentioned and explained in the Method (bandwidth of power extraction/maximal spectral density etc.)

Yes, the NLSE code has been run for different input pump powers.

Action taken

We added the section “**Dispersive wave generation efficiency**” to the Methods where we included more details on the theoretical evaluation of the CE:

“Also, we simulated the CE by integrating the output spectra obtained solving the NLSE. The CE was defined as the ratio between the integral performed over the DW bandwidth, over the integral of the input spectrum. The on-chip power was then calculated by multiplying the theoretical CE by the value of the pump power used in the simulation.”

Page 8-9: “Indeed increasing the pump power, up to our 20 mW maximum, does not result in a significant increase in DW power (see Fig. 4b). This could be due to the low values of 2β in these waveguides, leading to an already large soliton number (> 10) for coupled pump power in the 12-20 mW. This plateau is not observed in the two largest waveguides the soliton number remaining well below 10 for all tested coupled powers, and soliton-related dynamics being clearer.” => It would be useful to see this quantitative conversion efficiency from different aspect. Here, there is uniquely a consideration in term of power over a given bandwidth. How does this plateau behaves increasing/decreasing the power integration bandwidth ?

The integration bandwidth we used included the entire DWs bandwidths, and it was thus different from waveguide to waveguide. This is now clearly said in the new method section. Extending the bandwidth to include generation in other spectral region would not lead to a correct estimate of the DWG CE.

The soliton number clearly impact the peak power, compression point, bandwidth and breathing period(s) of the subsequent soliton fission in perturbed high-order soliton dynamics. There is probably more to dig in this respect (power and periodicity of that compression vs conversion efficiency), but the explanation is here a bit too qualitative in my opinion. Additionally, such a plateau seems to disappear when considering the simulation results, but are only limited by the available coupled power in the waveguides. Also, why are the authors defining $N=10$ as a threshold for this behaviour ? It seems rather arbitrary but is there another relevant element not disclosed ?

The plateau in Fig.4b happens for the two smaller waveguides for a coupled pump power between 10 and 12 mW, where the CE slope significantly changes (being negative in the 11-18 mW range). Indeed, this is also clearly observed in simulation. In these two waveguides, higher pump power would more likely induce a MI based dynamics, so that DWG would not be the main broadening mechanism towards the Mid-IR. As correctly pointed out by the reviewer, this is closely related to the soliton number, which is affected by β_2 in this case.

Action taken

We now better explain the physical principle of this behavior and clarify the role of MI with respect to the soliton number. Also, together with the above mentioned *Opt. Express* **13**, 3989 (2005) (ref 46), we included *J. Opt. Soc. Am. B* **19**, 1961 (2002) (now reference 37) where the reader can find the theoretical treatment which defines the scaling laws of high order soliton compression, including the dependence of the soliton compression distance (page 9 of the new version):

“Interestingly, the CE for the two smaller waveguides starts decreasing beyond a coupled power of about 12 mW, a behavior confirmed by the simulations. Indeed, increasing the pump power up to our 20 mW maximum, does not result in a significant increase in DW power (see Fig. 4b). This could be due to the low values of $|\beta_2|$ in these waveguides, leading to a longer compression point than in the two larger waveguides (see Supplementary Information). Therefore, in narrower waveguides, spectral broadening caused by modulation instability⁴⁶ can happen on the same length scale of soliton compression already for coupled pump powers in the 12-20 mW range. In fact, at a given point on the waveguide, modulation instability scales exponentially with the pump power. Equivalently, one can look at the soliton number, which also increases decreasing $|\beta_2|$. In fact, it has been shown that soliton fission happens before wave breaking by modulation instability becomes relevant when $N < 10^{34}$ ”

Page 8-9: “This plateau is not observed in the two largest waveguides the soliton number remaining well below 10

for all tested coupled powers, and soliton-related dynamics being clearer" => This sentence is not clear in my opinion and should be reformulated. What are clearer soliton-related dynamics ??

We removed the sentence soliton-related dynamics and rephrased the text as reported in the previous answer. In the new version, we explicitly mention that for larger pump power, MI competes with soliton fission and DWG as broadening mechanism for the narrower waveguides.

Methods – Numerical Simulations: I would recommend succinctly detailing the model (GNLSE ?) and provide the variables used (dispersion, nonlinearity, self-steepening ? Raman ? etc.). Is 110 fs the chirped pulse duration ? Maybe giving the bandwidth or transform-limited duration would help the reader.

In the revised version of the manuscript we included the requested information.

Action taken

We provided further details on the parameters used in NLSE in the Methods section “Numerical simulations”:

“We included the spectral dependence of the nonlinear coefficient and a full and complete dispersion profile that comprises up to 30th order when Taylor expanded with respect to the pumping frequency. Including more than 30 HOD terms didn’t change the simulation result and just slowed down the computation.”

We provide additional information on the pump source like average power, central wavelength, bandwidth and transform limited pulse duration at the beginning of page 7:

“The pump source is a commercial, turnkey soliton self-frequency shifted thulium-doped fiber mode locked laser (NOVAE Brevity $\lambda+$) with pulse duration at full width half maximum (FWHM) of 78 fs, bandwidth of 60 nm centered at 2090 nm, repetition rate of 19 MHz and average power of about 100 mW.”

Methods – Absorption spectroscopy: “We used 0.1 and 0.02 sampling interval for Fig. 4a and Fig. 4b respectively.” => I do not understand if Figs 4 a and b represent the same data or refer to different measurement. How are these data expected to be different?

There was an error in the manuscript, as the Figures to be considered were 5a and b (not Fig.4a and Fig.4b). We apologize for this typo and thank the referee for noticing it. Referring to Figure 5, the spectra in (a) represent the SC output in the region of interest after the cell with and without C₂H₂. In the second case, a vacuum pump was used to attenuate water absorption features inside the cell. In Fig.5b we report the normalized C₂H₂ spectrum to the reference (gas cell purged with pure N₂).

Action taken

We moved and slightly expanded these details to the Method section for better understanding:

“The sample spectrum is obtained by filling the gas cell with 396 ppm C₂H₂ buffered in N₂ in 1 atm total pressure at T=296 K. The reference is obtained by filling the gas cell with pure N₂”

Supplementary: A color scale on the density data plot would be appreciated.

We added it in the new version

The references contain several typos:

[30] Second author is John Roy Taylor

[45] Lowercase “A”

[46] “Si3N4”

[47] Journal name is missing

We thank the reviewer and we checked the references

Reviewer #2 (Remarks to the Author):

The authors demonstrate supercontinuum generation in a silicon nitride waveguide. The main novelty here is that they generate an asymmetric, powerful dispersive wave rather far from the pump through dispersive wave generation. This system is pumped by a fiber based system which makes the whole setup relatively simple. In my opinion the technical quality of the paper is really high. The measurements are done in a reproducible manner, the supercontinuum is measured as function of different parameters: the power, the waveguide width,... However, the main issue I have is that the novelty is rather limited. I leave it open to the editor to make a decision to evaluate this, but I would like to make sure that the editor takes the following points in account:

-The waveguides discussed here on Figure 1 are the same (at least the figures are exactly the same) as in the Nat Phot paper published by the authors in 2018. Although the pump wavelength is different the authors do a very similar experiment there. In fact the supplementary information of that paper includes some simulations indicating that pumping such a large cross section waveguide at a longer waveguide is interesting since the dispersive wave shifts further into the mid-infrared.

We thank the reviewer for acknowledging the quality and consistency of our measurements. In this work, we refer to the results reported in *Nat. Photonics* **12**, 330–335 (2018), where generation of Mid-IR frequency comb was demonstrated from a silica Er: doped silica fiber laser through a SiN chip waveguide. With this new work, we moved forward to tackle and eventually overcome the main limitation from this previous work: the extremely small conversion efficiency and power obtained in the Mid-IR. However we would like to point out that *none* of the experimental or numerical results showed in the *Nat. Photonics* **12**, 330–335 (2018) paper considered pumping wavelength beyond the telecom region. In the main text and supplementary information of *Nat. Photonics* **12**, 330–335 (2018) it was reported a pump (always at 1560 nm) to mid-IR conversion efficiency of less than 1% for DW emission beyond 3.2 μm . Please, note that supplementary Figure S2 of that paper refers to the numerical simulation of the pump to mid-IR conversion efficiency as a function of the central wavelength of the generated dispersive wave. Different waveguide dimensions and polarizations are reported, but all data refer to a pumping wavelength at 1560 nm.

Indeed, the main challenge of this work was to keep the same scheme, namely a silica fiber comb source + photonic integrated waveguide, to preserve the technological relevance of the above mentioned result and, at the same time, demonstrate its viability for actual spectroscopic application. As a matter of fact, other few platforms has been recently investigated following this paradigm, but none of them was actually able do demonstrate a spectroscopy experiment. In the new version of the introduction we provide a better comparison with such works and explain why our scheme is, up to know, the best candidate to realize an on-chip Mid-IR spectroscopic source.

-The idea to have a dispersive wave to extend a supercontinuum is not new, it has been shown on many integrated platforms and also in photonic crystal fibers and chalcogenide fibers. These are typically asymmetric because the group velocity dispersion of these fibers is very asymmetric around the pump.

The reviewer is right in saying that DWG was also reported before and we report the most relevant results in the introduction. However, we would like to point out that in fibers, the physics underlying is different, as Intrapulse Raman Scattering (IRS) plays a crucial role in DW dynamics too. Also, IRS in such works is the main responsible for the asymmetry of the supercontinuum generation, while in our case the effect of dispersion is the most relevant. Also, we would like to point out that in this work we selectively enhance the emission in a well-defined spectral region just by the use of waveguide engineering (dispersion and losses). To the best of our knowledge, this was never experimentally demonstrated before.

Action taken

We now better address this point including two new references (*Nat. Photonics* **1**, 653–657 (2007) and *Opt. Express* **21**, 11215 (2013)) at page 8 of the revised version

“In fact, because of the negligible Raman response³⁹, here we don’t observe the red-shift of the soliton and the consequent trapping of DW, as observed in silica fibers^{45,44}. ”

-In fact, it would make more sense to pump a (chalcogenide, InF₃,...) fiber with the source than an integrated waveguide. This would make sense in terms of coupling losses handling ... Such a product (2 um laser pumping InF) is commercial (https://www.thorlabs.com/newgrouppage9.cfm?objectgroup_id=10819) and has an efficiency of ~35%. A question arising is what can be added to this approach with an integrated solution?

It is true that the reported losses (about 5 dB per facet) are high. However, this problem can be solved by proper engineering of the inverse taper mode converters. Usually less than 3 dB losses per facet is perfectly achievable at the laboratory level in the telecom band, where a more mature design allows better mode size converter (*IEEE J. Sel. Top. Quantum Electron.* **24**, 1–11 (2018)). The coupling losses of 1.3 dB reported in *Opt. Express* **23**, 30592 (2015), the paper related to the commercial product mentioned by the reviewer, would definitively be not far from the values achievable by packaging photonic integrated circuits (*Appl. Sci.* **6**, 426 (2016)).

Also, not all the Mid-IR fiber platforms have better coupling losses handling. Mid-IR DW generated in nano-spike using a similar platform shows about 9 dB of input losses at best (Granzow, N. *et al. Opt. Express* **21**, 10969 (2013)).

Beyond obvious benefit in miniaturization and compactness, the use of the integrated platform and scheme demonstrated in our paper has also other benefits with respect to https://www.thorlabs.com/newgrouppage9.cfm?objectgroup_id=10819 (or the related publication *Opt. Express* **23**, 30592 (2015)):

- We have a better power scaling. In fact, thanks to the higher nonlinearity we don’t need a high power laser. In their experiment, the authors report a best CE similar to our result (35%) employing an high power Thulium doped fiber laser at 1969 nm with 400 mW average power, corresponding to 28 kW peak (3 nJ pulse energy), almost 4 times the one we couple to our waveguide.
- The long edge of the supercontinuum generated in InF₃ is most likely due to Raman shift. The authors didn’t assess experimentally the coherence of the Mid-IR part, but simulations suggest a degradation of the coherence increasing the length, which in turns leads to a trade-off between coherence and extension (power density) of the Mid-IR spectrum. In particular the reported efficiency was estimated in the longest fiber, with a clear loss of coherence.
- Our approach allows tuning the emission wavelength of the Mid-IR wave just by changing the waveguide width. This is not possible in the InF₃ fiber, where Mid-IR SCG was generated just for one specific fiber, therefore with fixed Mid-IR coverage. Also, in our Si₃N₄ waveguides, we can control the dispersion (and thus the emission wavelength) with lithographic precision, while this is definitively not possible in the InF₃ fiber.

Action taken

In order to better set our work in the context of Mid-IR generation in integrated platforms using silica fiber lasers (including both SCG and intrapulse DFG), we modified the introduction in this way:

“Broadband mid-IR generation can also be achieved by supercontinuum generation (SCG) in soft glass optical fibers^{14–23} or photonic integrated waveguides^{24–31}. Compared to other wavelength conversion schemes, SCG offers some advantages³²: it is in fact a compact single pass geometry that does not require any additional seed laser or temporal synchronization, and it can provide a broader and tunable mid-IR emission. When CMOS compatible materials are employed, SCG platforms benefit from lithographic precision and high yield, and usually have low power consumption. Recent works have shown that such approach can be applied as well to dual-comb spectroscopy³².

Often, to extend the reach into the mid-IR, the nonlinear waveguides are pumped with an optical parametric oscillator (OPO) placed beyond the 2 micron wavelength range, but there is interest in driving such platforms with femtosecond mode-locked fiber lasers, which are reliable, easy to use and compact frequency comb sources.

However, up to now, very few demonstrations succeeded in generating supercontinua following this paradigm. High energy (2-3 nJ) femtosecond pulses in the 2 micron region can lead to more than 30% of conversion efficiency (CE) beyond 2200 nm in a InF_3 fiber²², and spectral broadening extending beyond 4.5 μm in Telluride photonic crystal fibers²³. Despite the large CE and spectral extent, the coherence of such mid-IR emission has not been experimentally investigated. Conversely, low pulse energy (18 pJ) at 2000 nm can lead to coherent mid-IR SCG in ChG nanospikes^{16,17} or nanotapers¹⁹, but early damage power threshold and high mid-IR absorption from the cladding avoid power scaling up to the mW level. In terms of chip integrated waveguides, mid-IR extended SCG has been experimentally demonstrated in AlN waveguides pumped with 0.8 nJ from a telecom band femtosecond mode-locked fiber laser²⁴. The power generated inside the waveguide in the 3000 nm – 4000 nm range is about 0.3 mW, corresponding to a CE close to 0.5 %. Overall, none of the above-mentioned approaches has been employed for mid-IR spectroscopy demonstrations.

Also, in the last years there has been a trend in miniaturizing and simplifying DFG schemes by utilizing single pump configuration with chip-scale nonlinear platforms. Emission up to 5.5 μm was obtained by DFG between the pump and the long-wavelength dispersive wave (DW) in AlN waveguides²⁴, while intrapulse DFG, generated from an Er:doped fiber mode-locked laser, enables tunable mid-IR radiation in the 4-5 μm region using PPLN waveguides³³. However, efficiencies are limited to below 0.5 %.

Recently, we showed that direct generation of mid-IR light, from an erbium-doped fiber laser at 1.56 μm , is possible through DW generated in Si_3N_4 , and asserted the phase coherence and a frequency comb nature³⁴. This platform has the potential to merge all the desired features addressed separately in the above-mentioned devices: a power scalable, fiber laser pumped coherent mid-IR generation in a low loss chip-scale waveguide with lithographical control of its dispersion.”

Some additional short questions:

-The authors argue that they reach the greenhouse gas spectral region. CO₂ absorbs at 4.3 μm (not/barely reached in the paper) and the gas that is used in the gas cell (C₂H₂) is after a quick search not really considered an important greenhouse gas. I think the authors should remove the reference to the greenhouse gasses or make it clear what is meant here.

Indeed, the 3-4 micron region hosts absorption features of important greenhouse gases like Methane (CH₄), around 3.3 μm , and Nitrous oxide (N₂O) at 3.9 μm . As for CO₂, considering the expected low absorption losses of the large cross section SiN waveguides (see the new Fig.1b), we think that DW emission at 4.3 μm will be still possible with our approach, provided the correct waveguide dimension is employed. Moreover, we would like to point out that CO₂ has an absorption feature at 2.7 μm too, which is achievable, even with higher efficiency, with our narrower sample. Alternatively, some flexibility in the DW central wavelength can be obtained by tuning the wavelength of the pump source within the emission range of Thulium. Therefore we think that referencing to greenhouse gases as a possible application of our result is correct.

However, it is true that C₂H₂ is not a greenhouse gas, and it was only used to demonstrate that our DWG-based scheme is suitable for absorption spectroscopy experiments, based on availability of gases at the moment of the experiment. We agree that this distinction might not be clear enough to the reader in the abstract.

Action taken

We therefore removed the reference to greenhouse gases in the abstract, and the sentence is now rephrased in the following way:

“Here, we experimentally demonstrate that large cross-section Si_3N_4 waveguides pumped with 2 μm fs-fiber laser can reach the important spectroscopic spectral region in the 3-4 μm range, with up to 35% power conversion and milliwatt-level output powers.”

Instead, we still mention the greenhouse gas detection in the discussion, where we explicitly list the gases which could be target with our approach.

“The device can reach the 3-4 μm region, which hosts the signature of important greenhouse gases like Methane (CH_4) and Nitrous oxide (N_2O). The absorption lines of Carbon dioxide (CO_2), just at the boundaries of the covered range (around 2.7 μm and 4.3 μm), can be targeted with slightly narrower or wider waveguides. In addition, this approach provides a power level sufficient for spectroscopy application¹³, bridging the gap between fiber sources and quantum cascade lasers, which are the workhorse of mid-IR spectroscopy devices”

Also, the general wording “greenhouse gases” was added to the introduction, to include non-hydrocarbons greenhouse gases too.

Please, note that the gas absorption lines we are referring to can be seen at: <http://www.spectraplot.com/absorption>.

-From the paper it is not entirely clear how the conversion efficiency is calculated and over which band is integrated.

We thank the reviewer for helping in improving the completeness of our manuscript.

Action taken

In order to better specify this point, we included an additional section in the Methods, called “Dispersive wave generation efficiency”, where we detail the experimental and numerical procedure used for evaluating the SWIR to Mid-IR wavelength conversion efficiency:

“We estimate the DW power by integration on the FT-OSA. To first calibrate the power measurement in the FT-OSA, we directly coupled the attenuated pump laser at 2090 nm to the MMF via the parabolic mirror and sent it to the spectrometer. We measured the value obtained integrating the entire laser bandwidth in the FT-OSA ($P_{\text{FT-OSA}}$) and we compared it to the power detected with an InGaAs photodiode (Thorlabs S148C) at the output of the MMF (P_{PD}). The quantity $c = P_{\text{PD}}/P_{\text{FT-OSA}}$ gave us the calibration factor for the spectrometer. The mid-IR DWs were then integrated over their entire spectral extend, namely 86-113 THz, 80-110 THz, 76-105 THz and 70-90 THz for the 1000 nm, 1050 nm, 1100 nm and 1175 nm wide waveguides, respectively. These values were then multiplied by the measured calibration factor (c) at the pump wavelength to retrieve the mid-IR DW power at the MMF output. It is important to notice that c is constant over the entire FT-OSA spectral range. Finally, we considered the total out-coupling losses, including transmission through the MMF and the output lens, which was optimized for mid-IR throughput. The on-chip CE is then calculated as the ratio between the on-chip mid-IR DW power over the coupled pump power. The coupled pump power is estimated by direct detection of the pump laser before the chip, and taking into account the in-coupling losses from the input lens (5 dB). Also, we simulated the CE by integrating the output spectra obtained solving the NLSE. The CE was defined as the ratio between the integral performed over the DW bandwidth, over the integral of the input spectrum. The on-chip power was then calculated by multiplying the theoretical CE by the value of the pump power used in the simulation.”

-Is the coupling efficiency over the whole band flat. I would expect that it is very wavelength dependent.

The lens coupling efficiency is rather flat across the MLL band (about 60 nm), that is the entire input spectral range. However, mainly due to chromatic dispersion of the coupling lens, the coupling efficiency is not flat at all over the entire supercontinuum spectrum at the output. Therefore we optimized the focal distance of the output lens to maximize the Mid-IR DW throughput to estimate the Mid-IR DW power. Analogously, we set the focal distance to maximize the output at the pump wavelength in order to estimate the coupling losses. As we could detect the output power more precisely at the pump wavelength with a semiconductor detector, we assumed the outcoupling losses in the Mid-IR to be the same of that at 2090 nm, when the focal distance is optimized accordingly to the coupled wavelength.

Action taken

This is now clearly written in the new Methods sections, reported in the previous answer. Please notice that the coupled pump power used to retrieve the CE was estimated from in-coupling losses and therefore does not suffer from chromatic dispersion or differences in transmission across the input bandwidth.

Reviewer #3 (Remarks to the Author):

The paper claims that they have dispersion engineered SiN waveguides to achieve record high conversion efficiencies in generating light in the 3-4 micron region. This is a very important area for spectroscopy and the reported efficiencies are impressive.

In general I would recommend publication, but have a few comments to address below.

The authors mention high yield, it would be interesting to know the actual yield numbers.

The photonic damascene process ensures a sample yield of more than 95%.

Action taken

This information is reported in reference 36 of the new version of the manuscript.

Page 9 2nd paragraph: It is unclear in this paragraph if 1 mW average power was actually experimentally measured at the output of the chip or if the authors are simply pointing out that it could be achieved.

We measured the Mid-IR DW in the FT-OSA, as we didn't have a suitable detector for that wavelength range. After calibration of the FT-OSA (as explained in the new Methods section), we retrieve a power, just in the Mid-IR DW bandwidth, of about 1 mW at the chip out. Considering the out-coupling losses from the lens, this corresponds to the on-chip power reported in Fig.4b.

Action taken

In the manuscript, we changed the word "retrieved" with "estimated" (which might better fit our experimental procedure). Please, notice that now the mentioned sentence is at page 10, second paragraph. Also, we refer to Methods section for detailed information on the calibration of the FT-OSA, where we included the following:

"We estimate the DW power by integration on the FT-OSA. To first calibrate the power measurement in the FT-OSA, we directly coupled the attenuated pump laser at 2090 nm to the MMF via the parabolic mirror and sent it to the spectrometer. We measured the value obtained integrating the entire laser bandwidth in the FT-OSA (P_{FT-OSA}) and we compared it to the power detected with an InGaAs photodiode (Thorlabs S148C) at the output of the MMF (P_{PD}). The quantity $c = P_{PD}/P_{FT-OSA}$ gave us the calibration factor for the spectrometer. The mid-IR DWs were then integrated over their entire spectral extend, namely 86-113 THz, 80-110 THz, 76-105 THz and 70-90 THz for the 1000 nm, 1050 nm, 1100 nm and 1175 nm wide waveguides, respectively. These values were then multiplied by the measured calibration factor (c) at the pump wavelength to retrieve the mid-IR DW power at the MMF output. It is important to notice that c is constant over the entire FT-OSA spectral range. Finally, we considered the total out-coupling losses, including transmission through the MMF and the output lens, which was optimized for mid-IR throughput. The on-chip CE is then calculated as the ratio between the on-chip mid-IR DW power over the coupled pump power. The coupled pump power is estimated by direct detection of the pump laser before the chip, and taking into account the in-coupling losses from the input lens (5 dB). Also, we simulated the CE by integrating the output spectra obtained solving the NLSE. The CE was defined as the ratio between the integral performed over the DW bandwidth, over the integral of the input spectrum. The on-chip power was then calculated by multiplying the theoretical CE by the value of the pump power used in the simulation."

In Figure 5c, the zoom shows the observed absorption vs the HITRAN absorbance. I would expect the measured

absorption to have symmetric peaks similar to the HITRAN data, but they are not. The authors should provide an explanation for this and any impact it has on the validity of their measurement.

The mismatch between our measurement and the fit of the HITRAN data is quantitatively shown in the residual plot. Overall, the matching is good except of some spikes which appear between 3000 nm and 3100 nm. We run additional simulations to better understand this feature and we came to the conclusion that it is due to a deviation from linearity in the wavelength accuracy of our spectrometer. In fact, this is the only parameter not taken into account in the HITRAN fit, where we consider the same wavelength correction for all the absorption lines. Also, we observed the same spikes in the residuals setting in the fit the same wavelength resolution used in the experiment. This excludes the limited wavelength resolution of our OSA as the cause of this feature.

Action taken

We modified the last sentence of the section “Proof-of-principle spectroscopy measurement of C₂H₂” in the following way:

“Clear spikes in the residuals near the absorption peaks are likely due to the wavelength nonlinearity of the OSA within the detection range, which is not taken into account in the model.”

In the Discussion section the authors claim that they reach output power levels at 4μm which are comparable to the state-of-the-art mid-IR sources for dual comb spectroscopy, and that they are pumped with a similar fiber frequency comb. However, the repetition rate of the comb used in this paper is only 19 MHz, and the paper from (12) is at 200 MHz. They go on to say that they can obtain similar output power levels with roughly ten times less average pump power than (12). I believe this comment to be a bit misleading, as they are using a source that is roughly 10x less in repetition rate. The nonlinearity goes by peak power in the pulse, so if the authors switched to a 200 MHz source (in order to be similar with ref 12, they would require 10x more average power to achieve the same level of broadening. The comparison should be better explained.

In the mentioned reference (reference 13 in the new version), the authors consider as pump for the DFG the near-IR source at around 1 μm obtained from SCG in a HNLF pumped with a telecom-band fiber comb at 200 MHz. The signal for the DFG is a SCG from 1350 nm to 1750 nm (pumped with the same comb) which is not dispersion compensated and we therefore assume with lower peak power (average power is 450 mW). Let’s thus just consider the pump power for efficiency comparison.

In reference 13, when using PPLN bulk crystals for DFG, the pump duration is 200 fs with average power 900 mW. This corresponds to a peak power of 22.5 kW which is from 2 to 4 times larger than the peak power we couple into the waveguide. Also, in reference 13, the authors used a chirped PLN waveguide with a stretched 1 ps pulse to generate broadband DFG. The peak power is in this case around 4.5 kW, lower than the one we use (which is from 5 to 10 kW, depending on the waveguide).

Summarizing, we can thus say that the pump laser employed in reference 13 has roughly the same peak power we couple in the chip with ten times higher repetition rate, this leads to the factor of 10, referred to the average power and mentioned in the first version of the text.

The generated Mid-IR average power in reference 13 strongly depends on the emission wavelength, and varies from 1 mW to 100 mW. Therefore, even if it might be correct to consider the emitted power comparable at 4 μm, on average we can consider it is tens of milliwatt, about 10 times higher than what we report. However, this is a consequence of the ten times higher repetition rate employed in reference 13, while indeed the *energy* emitted in the Mid-IR is comparable in the two approaches. In practice if we have had a pump source with same peak power and repetition rate of reference 13, we would have had the same average Mid-IR power.

Action taken

We agree with the reviewer that it makes more sense to speak about coupled *peak* power. To avoid any confusion we eliminate the sentence containing the input average power claim and to better compare our result over the entire Mid-IR range investigate, we rephrased the sentence in the following way:

“Notably, the mid-IR output energy we obtained is comparable to the state-of-the-art mid-IR sources for dual comb spectroscopy, based on DFG in PPLN waveguides¹³, pumped with similar peak power fiber frequency comb.”

The authors claim that their source could be used for dual-comb spectroscopy, but a 19 MHz source would not be useful for this as the comb modes are spaced too close together limiting the total optical bandwidth that could be sampled and the power per comb mode would be very low.

I suggest revising the claims made in the Discussion section per the comments above in order to make a more realistic comparison.

We thank the reviewer for giving us the opportunity to better clarify this point, in view of future applications of our scheme.

As for the SCG dynamics, the repetition rate does not affect the DWG generation process and its bandwidth, as long as the peak power of the pulse is sufficient. For example the telecom band MLL used in *Nat. Photonics* **12**, 330–335 (2018) featured 100 MHz of repetition rate but showed similar dynamics having similar kW peak power coupled in the chip. In general, the lowering in peak power which could derive from an increase in the repetition rate can be compensated by better optimization of the input mode size converter and recompression of the input pulse after polarization optics.

As for comb spectroscopy, close teeth spacing can be employed in narrow line spectroscopy, e.g. for detecting isotopes. However, it is true that in dual comb spectroscopy (DCS), the maximum optical bandwidth (BW) which can be probed depends on the square of the repetition rate (f_r) through the relation $BW \leq f_r^2/(2\Delta f_r)$ (*Optica* **3**, 414 (2016)).

Typical values of Δf_r spans from 100 Hz to 1 kHz (*Optica* **3**, 414 (2016)), but DCS based on fiber laser pumped DFG has also been demonstrated with $\Delta f_r = 50$ Hz, see for example (*Nat. Photonics* **12**, 202–208 (2018), *Opt. Lett.* **43**, 1678–1681 (2018) and *Optica* **5**, 727 (2018)). With such a value for Δf_r , an optical bandwidth up to about 4 THz can be probed by DCS with the current source. Considering a center wavelength at 3050 nm, like in the spectroscopy experiment reported in our work, we could thus probe about 120 nm, enough to include all the acetylene absorption lines. This value would obviously improve at longer wavelengths. In fact at 4 μm we would have $BW > 200$ nm.

Moreover, the 2 micron fiber laser technology is improving very fast and the repetition rate of these lasers can now scale up to GHz level, preserving hundreds of femtoseconds of pulse duration (*Opt. Express* **26**, 24687 (2018)). Also, sub-100 fs fiber lasers are commercially available in the two micron region showing higher repetition rates, and tens of kW of peak power. For example Thorlabs FSL1950F has $f_r = 50$ MHz while $f_r = 115$ MHz has been reported in *Nat. Photonics* **12**, 209–214 (2018) using a laser from IMRA America. In this case, the DCS bandwidth dramatically increases up to 100 THz ($\Delta f_r = 50$ Hz), corresponding to more than 3 μm at 3050 nm. Alternatively, almost 400 nm at 3050 nm (corresponding to 760 nm at 4 μm) can be reached with $\Delta f_r = 400$ Hz, allowing to a reduction of the measurement time down to 2.5 ms for line resolved experiment. Moreover, in literature (see for example: *Opt. Express* **23**, 26596 (2015)), Tm fiber lasers with f_r more than 400 MHz pulse duration ~ 100 fs and tens of kW of peak power are reported.

We agree with the reviewer in saying that, using a higher repetition rate laser with similar coupled pump power, the power per comb mode would be proportionally larger.

Action taken

We therefore changed the following sentences in the discussion part:

“The expected comb structure of the DW³³ could also enable dual comb measurements, which however would be best implemented pumping the waveguides using a laser with hundreds of MHz of repetition rate⁸.”

And

“Such result could therefore provide a suitable alternative to microresonators⁴⁸⁻⁵² to generate mid-IR frequency combs on a chip, when sub-GHz teeth spacing is required.”

REVIEWERS' COMMENTS:

Reviewer #1 (Remarks to the Author):

Following my previous assessment on the manuscript 'Mid infrared gas spectroscopy using efficient fibre laser driven photonic chip-based supercontinuum', the authors have taken significant actions in improving their manuscript and have addressed my comments. I thus gratefully thank the authors for their careful and detailed reply.

As per my initial review, I still think that the proof-of-concept demonstration of spectroscopy is significant, the paper is interesting and well-written, and the results are valid. In my view, the overall conceptual novelty is still somehow limited. However, given the quality of the results (e.g. in terms of conversion efficiency) and the potential impact of the study (as e.g. demonstrated by spectroscopic applications), I would recommend publication of the manuscript, in its current state, in Nature Communications.

Reviewer #2 (Remarks to the Author):

The reply of the reviewers is impressive. They have taking in account all my comments and addressed them very well. I am in favor of publication

Reviewer #3 (Remarks to the Author):

The authors addressed my suggestions in full. The article is novel, well written, technically sound, and interesting to the field, therefore, I am happy to recommend publication.